# Downwelling surface solar irradiance in the tropical Atlantic Ocean: A comparison of re-analyses and satellite-derived data sets to PIRATA measurements

Mélodie Trolliet[1], Jakub P. Walawender[2], Bernard Bourlès[3], Alexandre Boilley[4], Jörg Trentmann[2], Philippe Blanc[1], Mireille Lefèvre[1], Lucien Wald[1]

[1] MINES ParisTech, PSL Research University, O.I.E. - Center for Observation, Impacts, Energy, Sophia Antipolis, France
[2] Deutscher Wetterdienst, Offenbach, Germany
[3] IRD/LEGOS, Brest, France
[4] Transvalor, Mougins, France

*Correspondence to*: Mélodie Trolliet (melodie.trolliet@mines-paristech.fr)

Running title: M. Trolliet et al.: Comparing PIRATA measurements, re-analyses and satellite-derived data.

**Abstract.**

This paper assesses the merits and drawbacks of several data sets of solar downwelling radiation received at the horizontal surface of the tropical Atlantic Ocean where the magnitude of this radiation and its spatial and temporal variability are not well known. The data sets are compared to quality-controlled measurements of hourly means of irradiance made at five buoys of the PIRATA network for the period 2012-2013. The data sets comprise the re-analyses MERRA-2 and ERA5, and three satellite-derived data sets: HelioClim-3v5, SARAH-2 and CAMS Radiation Service v2. It was found that the re-analyses MERRA-2 and ERA5 often report cloud-free conditions while the actual conditions are cloudy, yielding an overestimation of the irradiance in such cases, and reciprocally, they report actual cloud-free conditions as cloudy, yielding an underestimation. The re-analyses exhibit more bias in irradiance in cases of medium and high level clouds than for low level clouds. They correlate well with the hourly means of irradiance (as a whole, correlation coefficients greater than 0.85 for MERRA-2 and 0.89 for ERA5); they correlate very poorly with daily means of irradiance (coefficients less than 0.48 and 0.59 for MERRA-2 and ERA5 respectively) and with the hourly and daily clearness indices (coefficients less than 0.53 and 0.46 for MERRA-2 and less than 0.63 and 0.59 for ERA5). The irradiance pattern at both hourly and daily timescales is spatially distorted by re-analyses, especially for MERRA-2. The three satellite-derived data sets exhibit similar performances between them. The correlation coefficients are greater than 0.95 and 0.78 for irradiance and clearness index respectively in most cases for hourly values and 0.90 and 0.88 for daily values. The relative standard deviation of errors is of order of 15 % for hourly values and 8 % for daily values. It is concluded that these data sets reproduce well the dynamics of the irradiance and clearness index at both hourly and daily timescales. They exhibit overestimation, with the lowest biases reached by CAMS Radiation Service v2 and ranging between 11 and 37 W m⁻² depending on the buoy. It is suggested that

HelioClim-3v5 and CAMS Radiation Service v2 are suited for reproducing the spatial gradients of the irradiance and reflecting the spatial variability of the irradiance.

## 1. Introduction

Solar radiation reaching the ocean surface is an essential variable in the ocean-climate system (Budyko, 1969; Manabe, 1969; Siegel et al., 1995; Lean and Rind, 1998). The density of power received from the sun on a horizontal surface at sea level per unit surface is called the downwelling solar irradiance at the surface and is hereafter abbreviated as DSIS. Other terms may be found in literature, such as solar exposure, solar insolation, solar flux, surface solar irradiance, downwelling shortwave flux or surface incoming shortwave radiation. The DSIS intensity is large over the tropical Atlantic Ocean and influences the sea surface temperature. The net downward surface energy is positive and accumulates within the ocean, resulting in a northward meridional transport of heat in the Atlantic Ocean (Liu et al., 2017). The DSIS influences the vertical structure of the ocean at more rapid timescales with local impacts on physics and plankton (Siegel et al., 1995).

Currently, the field of DSIS is not well known in the Atlantic Ocean. One of the means of assessing the DSIS is to use measuring stations such as pyranometers aboard ship or mounted on buoys (Cros et al., 2004). Such measurements are usually accurate. However, there are too few stations to offer a synoptic view of the DSIS field. Images acquired by satellites observing the ocean surface are a second means of getting a synoptic view of the temporal variations of the DSIS field. For example, the series of geostationary Meteosat satellites offers synoptic views of the tropical and equatorial Atlantic Ocean every 15 min with a spatial resolution of between 3 and 5 km. Several data sets of DSIS have been constructed from these images, such as the HelioClim-3, SARAH-2 (Surface Radiation Data Set – Heliosat, version 2), and CAMS (Copernicus Atmosphere Monitoring Service) Radiation Service v2 data sets which are dealt with in this paper (see Section 1 for further details).

Re-analyses are a third means. They derive from weather forecast models used in a re-analysis mode to reproduce what was actually observed. They assimilate state variables such as temperature, moisture and wind. In contrary to these variables, the DSIS is diagnostic i.e. it is derived from a radiative transfer model and depends on the representation of the whole set of radiatively active variables in the atmospheric column above the point. Hence, re-analysis estimates should not be mistaken with measurements of DSIS, because they include the uncertainty of the models. The data sets of interest here are the ERA5 developed at the ECMWF (European Center for Medium-range Weather Forecasts) and MERRA-2 (Modern-Era Retrospective Analysis for Research and Applications, version 2) of the NASA (National Aeronautics and Space Administration).

Despite the fairly recent availability of gridded data sets, their use is spreading outside the climate community. However, for a more informed usage of these data in ocean sciences as a whole, greater validation efforts are needed. This paper aims at establishing the merits and drawbacks of each of the five data sets when compared to quality-controlled hourly and daily measurements of DSIS recorded by the PIRATA (Prediction and Research Moored Array in the Tropical Atlantic) network

of moored buoys in the tropical Atlantic Ocean, here considered as a reference. The data sets are briefly presented in Section 1. The protocol for validation is presented in Section 2. The merits and drawbacks of each data set are discussed in Section 3. The size of the grid cell is typically 3 km for satellite-derived data sets and 50 km for the re-analysis data; it is large compared to a single point and this difference is discussed in Section 3. How to access the data is described in the "data availability" section.

## 2. Data-sets

### 2.1. The PIRATA measurements

The PIRATA network consist of eighteen meteo-oceanic buoys (ATLAS type, progressively replaced by T-FLEX systems from 2015) located in the Atlantic Ocean, between the latitudes 19° S and 21° N (Bourlès et al. 2008). Each PIRATA buoy is equipped with an Eppley pyranometer mounted at a height of 3.8 m that measures the DSIS. Values are recorded as 2 min averages. The sensors are deployed for about one year on average before replacement. Sensors are replaced with clean sensors during every yearly servicing cruise.

The measurements are subject to the same sources of uncertainty as their counterparts on firm ground, such as incorrect sensor levelling, shading caused by close structures, dust, dew, water-droplets, bird droppings, miscalibration of sensors, electronic failures, time shifts in data loggers, maintenance mishandling etc. (see e.g. Muneer and Fairooz, 2002). Some buoys experience accumulation of African dust which potentially leads to a significant underestimation of the DSIS. Foltz et al. (2013) have proposed corrections for the data from such buoys including corrections for sea-spray, natural and anthropogenic aerosols but limited to daily means of DSIS.

Pyranometers view a complete hemisphere and must be placed horizontally for accurate measurements. This is not the case within the PIRATA network where a pyranometer is affected by the motions of the buoy which change the portion of the sky seen, inducing errors in the measurements. The errors are very complex to estimate and correct (Katsaros and DeVault, 1986; MacWhorter and Weller, 1991). They depend on the relative sun-buoy geometry which may be expressed as the tilt angle, the angle between the plane of the pyranometer and the horizontal plane, and the difference in azimuth of the sun and tilt direction. This relative geometry is affected by wave action or strong surface current and depends on the time of the day, latitude and season. Since the downward radiation received from a portion of the sky depends on the sky conditions, the errors depend also on sky conditions. Errors are most apparent in conditions of high DSIS, in cloudless skies, and when solar zenithal angles are less than 60°. Katsaros and DeVault (1986) distinguished two main types of errors: those due to rocking motion caused by waves and those due to a mean tilt. The first type can be approached by the two following extreme cases: *(i)* the buoy motion is in the direction of the sun and *(ii)* the buoy motion is perpendicular to that direction. In the first situation, these authors expressed the error in irradiance measurement as a combination of losses produced by a motion away from the sun and gains by the tilting of the buoy toward the sun. By means of an analytical model and gross assumptions, they concluded that "the average error for a cycle of motion will not be zero but will not be large". In the second situation,

the effect of a perpendicular movement is always a loss, due to the loss of the sky portion seen by the pyranometer. Katsaros and DeVault (1986) calculated that the loss is of the order of 10 % in hourly mean of irradiance for 10° tilt and solar zenithal angle greater than 30°. For daily averages, the influence of the buoy movement is a combination of the two cases. As a consequence, compensating errors would often lead to smaller errors in measurement of daily means of irradiance. Wave action and a preferential tilt have the least effect in the tropics. However, diurnal variations in cloudiness, which are typical at low latitudes, make the compensating gains and losses uneven over the day, and therefore result in a larger net diurnal error than observed (Katsaros and DeVault, 1986). MacWhorter and Weller (1991) experimentally confirmed these calculations with simultaneous measurements of irradiance by gimbaled and un-gimballed pyranometers. Systematic tilts of 10° induced by strong surface currents or strong winds yield relative errors in excess of 40 %. Errors caused by wave motion are less severe and may amount to 10 %. Reynolds (2007) proposed an algorithm for correcting such errors. Inputs to this algorithm are the pitch, roll and heading of the sensor as well as the relative contributions of the beam and diffuse components of the DSIS. Long et al. (2010) suggested using a combination of a specific pyranometer and algorithm to achieve an accuracy of 10 W m$^{-2}$ in 90 % of the cases.

Currently, no correction is made to PIRATA measurements for errors due to buoy tilt or soiling. Measurements of 2 min DSIS for the period 2004-2016 were downloaded from the NOAA (National Oceanic and Atmospheric Administration) PMEL (Pacific Marine Environmental Laboratory) web site. Quality flags are provided together with the measurements. The NOAA procedure for quality checking rejects implausible values, i.e. values exceeding 1400 W m$^{-2}$. If any DSIS value, mean, standard deviation, or maximum, reads 0, all values are set to missing for that day. Flags are also raised if sensor outputs are zero or full scale throughout the day, or if the daily mean of the DSIS is outside the interval [50, 325] W m$^{-2}$. In a third pass, a visual inspection and comparison with time series plots from neighbouring sites are performed.

An additional quality control was performed at MINES ParisTech on top of the NOAA screening since the PIRATA measurements serve as a reference in this comparison. The quality control used here is that of Korany et al. (2016) and comprises several tests of the 2 min DSIS data against extremely rare limits and physically possible limits. Values falling outside these limits were excluded from the time-series. Eventually, a visual analysis was performed to further remove suspicious values. A noticeable fraction of the data was removed. Only measurements that successfully passed all tests were kept. The hourly mean of DSIS was computed by averaging the 30 measurements within the hour only if all measurements were declared valid. Otherwise, the hourly mean of DSIS was excluded.

The buoys located between 4° N and 8° N were discarded because of the possible large occurrence of significant tilt due to currents, and those located north of 8° N were discarded because of contamination by African dust (Foltz et al., 2013). A further constraint in this study was the availability of enough measurements at each buoy without major gaps in a year in order to have an accurate description of the intra-year variability. In addition, the period for which data sets overlap start in 2010 as ERA5 was only available for the period 2010-2016 at the time of writing.

Eventually, five buoys out of eighteen PIRATA buoys have been selected: (0°N, 0°E), (0°N, 10°W), (0°N, 23°W), (6°S, 10°W) and (19°S, 34°W) (Fig. 1). These buoys have enough hourly means of DSIS for the period 2012-2013 to guarantee

robust comparisons (Table 1). An additional ensemble of data was used to further controlling results and analyses: the year 2011 for the buoy (19°S, 34°W) which offered approximately 4200 measurements, and the years 2010 and 2011 for the buoys (0°N, 10°W) and (6°S, 10°W).

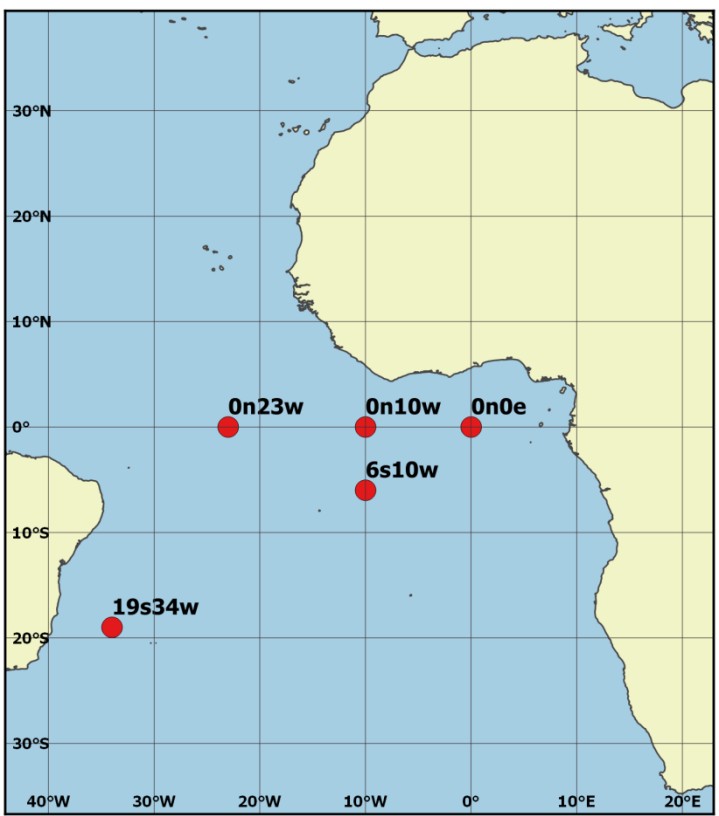

**Figure 1: Location of the five PIRATA buoys used in this study.**

| Buoy | Latitude (positive North) | Longitude (positive East) | Number of valid hourly values | Hourly mean of DSIS (W m⁻²) | Mean hourly clearness index | Daily mean of DSIS (W m⁻²) | Mean daily clearness index |
|---|---|---|---|---|---|---|---|
| (0°N, 0°E) | 0.0 | 0.0 | 8198 | 457 | 0.48 | 214 | 0.52 |
| (0°N, 10°W) | 0.0 | -10.0 | 8182 | 493 | 0.52 | 232 | 0.56 |
| (0°N, 23°W) | 0.0 | -23.0 | 7149 | 549 | 0.58 | 256 | 0.62 |
| (6°S, 10°W) | -6.0 | -10.0 | 8248 | 487 | 0.53 | 235 | 0.57 |
| (19°S, 34°W) | -19.0 | -34.0 | 8335 | 507 | 0.58 | 242 | 0.61 |

**Table 1: Geographical coordinates of the PIRATA buoys used in this study, number of hourly values in each time-series, hourly and daily means of valid data in DSIS and clearness index for the period 2012-2013.**

Table 1 reports the hourly and daily mean of the DSIS as well as the means of the hourly and daily clearness indices ($KT$). $KT$ characterizes the optical state of the atmosphere better than the DSIS and allows for comparing the modelling of the overall transparency of the atmosphere between models. Let $E$ be the hourly mean of the DSIS and $E0$ the corresponding irradiance received on a horizontal plane located at the top of the atmosphere. The hourly clearness index $KT$ is defined as

the ratio of $E$ to $E0$. $E0$ was computed here using the SG2 algorithm (Blanc and Wald, 2012). Though $KT$ is not completely independent of the position of the sun, the dependency of $KT$ on the solar zenithal angle is much less pronounced than that of $E$. $KT$ is an indicator of the optical transparency of the atmosphere; it is typically close to 0.8 in cloud-free conditions, and close to 0.1 in overcast conditions with optically thick clouds.

The daily mean of the DSIS was computed by summing up the hourly means and dividing by 24 h by convention. The daily

clearness index was computed in the same way as the hourly $KT$. The means of the daily $KT$ are greater than 0.5 denoting that the selected stations experience large occurrences of cloud-free conditions. Table 1 shows a tendency for an increase in $KT$ from east to west.

### 2.2.    The satellite-derived data set (HC3v5, SARAH-2, CRS)

The HelioClim-3 v5 and CAMS Radiation Service v2 data sets, abbreviated respectively as HC3v5 and CRS, are constructed

by processing images of the Meteosat Second Generation satellites by respectively the Heliosat-2 method (Rigollier et al., 2004; Lefèvre et al., 2007) modified by Qu et al. (2014), and the Heliosat 4 method (Qu et al., 2017). The SARAH-2 data record (Pfeifroth et al., 2017) is obtained thanks to a retrieval approach based on Heliosat-2 (Müller et al., 2015; Pfeifroth et al., 2018) exploiting the observations from Meteosat first and second generation. All data sets cover Europe, Africa, Middle East, parts of South America and Atlantic Ocean (full Meteosat disc).

HC3v5 and CRS are available from 2004 onwards with a 15 min time step. The spatial resolution depends on the pixel position and is approximately 3 km in the tropical Atlantic Ocean. Data can be accessed through a Web service at the SoDa Service (Gschwind et al., 2006, http://www.soda-pro.com/). This Web service performs integration over time. It delivers the DSIS in all sky conditions, the DSIS in cloud-free conditions, and $E0$. These three quantities were downloaded as hourly means and daily means for both HC3v5 and CRS for the locations of the selected buoys.

SARAH-2 is generated and distributed by EUMETSAT CM-SAF (Satellite Application Facility on Climate Monitoring). It provides information on the global and direct DSIS as well as the sunshine duration from 1983 to 2015. The data is provided on a regular grid with a grid spacing of 0.05° x 0.05° as instantaneous values of the DSIS every 30 min as well as daily and monthly averages. In this study, the instantaneous values every 30 min were converted into instantaneous clearness index every 30 min. The clearness index was resampled from 30 min to 1 min by temporal interpolation and then multiplied by the

corresponding $E0$ to yield a time series of 1 min irradiance. The hourly mean irradiance is the average of the 1 min irradiances over the full hour.

### 2.3. The re-analysis data sets (MERRA-2, ERA5)

The MERRA-2 data set has many of the same basic features as the MERRA system (Rienecker at al., 2011) that has already been validated against PIRATA daily means of DSIS by Boilley and Wald (2015), but includes a number of important updates (Gelaro et al., 2017). MERRA-2 offers 72 vertical levels of DSIS from the ground up to 0.01 hPa. The grid cell is 0.5° (approx. 55 km) in latitude by 0.625° (approx. 71.5 km at the Equator) in longitude. MERRA-2 offers hourly means of DSIS starting from 1980.

ERA5 is the fifth generation of ECMWF atmospheric re-analyses of the global climate (Hersbach and Dee, 2016). It has several improvements compared to ERA. It has 137 levels from the surface up to 0.01 hPa. The size of the grid cell is 31 km. At the time of writing, the temporal coverage is 2010 to the present with 1 h time step. The period will be extended back to 1979 at the beginning of 2018.

The hourly means of DSIS in all sky conditions and *E0* were downloaded from the MERRA web site for MERRA-2 and from the ECMWF MARS web site for ERA5. In addition, the DSIS under cloud-free conditions was downloaded for MERRA-2. The DSIS and *E0* time series for the location of each PIRATA buoy were constructed by firstly downloading the time series for the nearest four grid cells surrounding the buoy location, and then applying a spatial bilinear interpolation technique with a weighting factor that is inversely proportional to the distance to the PIRATA site. The daily means of DSIS were computed by summing up the hourly means and dividing by 24 h.

The characteristics of the five data sets are summarized in Table 2.

| Data set | | Start | End | Time resolution | Spatial resolution | Available from |
|---|---|---|---|---|---|---|
| Satellite-derived | HC3v5 | 2004 | present | 15 min | ~ 3 x 3 km² | SoDa Service |
| | SARAH-2 | 1983 | 2015 | 30 min | ~ 5 x 5 km² | EUMETSAT CM-SAF |
| | CRS | 2004 | present | 15 min | ~ 3 x 3 km² | SoDa Service |
| Re-analysis | MERRA-2 | 1980 | present | 1 h | 55 x 71.5 km² | NASA |
| | ERA5 | 2010 | present | 1h | 31 x 31 km² | ECMWF MARS |

**Table 2: Description of the data sets as on March 2018.**

### 2.4. The CAMS cloud classification

In addition to these data sets, other variables have been downloaded to support the analyses of the errors for each data set. The CAMS Radiation Service offers a detailed mode, so called the verbose mode, by which one may download several variables such as the fraction of pixel covered by cloud, solar zenithal angle and the aerosol optical properties and the following cloud types as a function of altitude (Qu et al., 2017):

- low level cloud: water cloud at low altitude, with a base height of 1.5 km and a thickness of 1 km;
- medium level cloud: water cloud at medium altitude, with a base height of 4 km and a thickness of 2 km;

- high level cloud: deep cloud of large vertical extent from low altitude to medium altitude, with a base height of 2 km and a thickness of 6 km;

- thin ice cloud: ice cloud with a base height of 9 km and a thickness of 0.5 km.

### 3. Protocol for comparisons

The present work followed the protocol that was designed and is used in the framework of CAMS to perform quarterly validation of the CRS products against qualified ground measurements (see reports by Lefèvre and Wald at https://atmosphere.copernicus.eu/validation-supplementary-products). It comprises two parts.

The first part consists in the computation of differences between estimates and measurements. These differences are then summarized by classical statistical quantities. In this part, one more constraint applies to the PIRATA measurements: any
measurement should be greater than a minimum significant value. This threshold is defined in such a way that there is a 99.7 % chance that the irradiance is significantly different from 0 and that it can be used for the comparison. It is set to 30 W m$^{-2}$, i.e. 1.5 times the uncertainty (percentile 95) of measurements of good quality as reported by the WMO (World Meteorological Organization, 2012). Otherwise, the measurement, and therefore the corresponding estimate, is not included in the computation of the differences. Following the ISO standard (1995), the differences are computed by subtracting
PIRATA measurements from the estimates. The set of differences is summarized by a few indicators namely the bias (mean of the differences), the standard deviation and the root mean square error. Relative bias, standard deviation and root mean square error are computed relative to the mean of the corresponding PIRATA measurements at a given buoy. Correlation coefficients are computed. 2D histograms of PIRATA measurements and estimates are drawn as well as histograms of the differences.

Statistical properties of estimates and measurements are compared in the second part. Histograms of both the PIRATA measurements and the estimates are computed, and are superimposed into a single graph. Such graphs aim at assessing the capability of a given data set to accurately reproduce the frequency distribution of the PIRATA measurements for the period. Monthly means and standard deviations within each calendar month of both the PIRATA measurements and the estimates are computed and displayed on a single graph.

In addition to the protocol for CRS validation, other graphs have been drawn to study the dependency of the statistical indicators on the irradiance or the clearness index, and other variables such as the month, year, solar zenithal angle, cloud types, cloud coverage, water vapour content, the aerosol optical properties or month. These graphs are not shown, except a few.

This protocol was applied to both $E$ and $KT$. $KT$ is less sensitive than $E$ to changes induced by the daily and seasonal effects
due to the sun. Hence, it is better indicator than $E$ to assess the performances of a model regarding its ability to estimate the optical state of the atmosphere. $KT$ was computed for each data set using $E0$ given by this data set to avoid artificial

distortions in results because of differences in *E0*. *E0* differs slightly from one data set to another by a few W m$^{-2}$, except for MERRA-2 as shown later.

The protocol was first applied to each data set for the five buoys for the period 2012-2013, using hourly values. In order to guarantee a better control and to support the conclusions, it was also applied to:

- the same data sample using daily values, with a threshold of 7.5 W m$^{-2}$ instead of 30 W m$^{-2}$,
- each data set for the buoys: (0°N, 10°W) and (6°S, 10°W) for 2010-2011, for both hourly and daily values,
- each data set for the buoy (19°S, 34°W) in 2011, for both hourly and daily values.

The study has been conducted also on daily values for several reasons. One reason is that the performances may differ across these different timescales. Another reason is that the daily values are the basis for calculating the monthly and yearly means,

which are used in climatology, and thus it is important to assess the quality of the data sets at the daily timescale. In addition, dealing with daily values allows comparing the present results to already published works (see Section 4 "Results and discussion" for details).

It was found that the results are similar between the three satellite-derived data sets on the one hand, and between the two reanalyses on the other hand. Hence, the description of the results and the discussion are organized as follows. The three

satellite-derived data sets are discussed altogether firstly for the hourly values, then for the daily values. Then, the two reanalyses are discussed for the hourly then daily values.

## 4. Results and discussion

### 4.1. The satellite-derived data sets (HC3v5, SARAH-2, CRS)

#### 4.1.1. Hourly values

Table 3 shows the correlation coefficients, the biases and the standard deviations at each buoy, for the three satellite-derived data sets, for hourly means of *E* and *KT* (correlation coefficient only) in the period 2012-2013. These quantities are similar to those for 2010 and 2011 (not presented).

| | Correlation coefficient | | | Bias | | | Standard deviation | | |
|---|---|---|---|---|---|---|---|---|---|
| Buoy | HC3v5 | SARAH-2 | CRS | HC3v5 | SARAH-2 | CRS | HC3v5 | SARAH-2 | CRS |
| (0°N, 0°E) | 0.964 | 0.969 | 0.966 | 48 | 56 | 31 | 79 | 72 | 75 |
| | *(0.854)* | *(0.869)* | *(0.878)* | (11 %) | (12 %) | (7 %) | (17 %) | (16 %) | (16 %) |
| (0°N, 10°W) | 0.951 | 0.963 | 0.956 | 57 | 57 | 37 | 92 | 79 | 86 |
| | *(0.780)* | *(0.825)* | *(0.817)* | (12 %) | (12 %) | (8 %) | (19 %) | (16 %) | (17 %) |
| (0°N, 23°W) | 0.969 | 0.975 | 0.972 | 32 | 28 | 18 | 73 | 66 | 68 |
| | *(0.816)* | *(0.841)* | *(0.834)* | (6 %) | (5 %) | (3 %) | (13 %) | (12 %) | (12 %) |
| (6°S, 10°W) | 0.966 | 0.978 | 0.968 | 42 | 43 | 29 | 77 | 62 | 74 |
| | *(0.851)* | *(0.890)* | *(0.880)* | (9 %) | (9 %) | (6 %) | (15 %) | (12 %) | (15 %) |
| (19°S, 34°W) | 0.927 | 0.923 | 0.929 | 33 | 12 | 11 | 117 | 119 | 113 |
| | *(0.642)* | *(0.626)* | *(0.661)* | (6 %) | (2 %) | (2 %) | (23 %) | (23 %) | (22 %) |

**Table 3: For each PIRATA buoy and each satellite-derived data set: Correlation coefficient between measurements and estimates from satellite-derived data sets for irradiance and clearness index *(in italics),* bias and standard deviation (W m$^{-2}$) between measurements and estimates for irradiance with relative values (in brackets) for hourly means in the period 2012-2013.**

5   The three estimates correlate very well with the measurements. The correlation coefficients are very similar between data sets for both *E* and *KT*. They range between 0.92 and 0.98 for *E* (Table 3). They are slightly less for *KT* and range between 0.79 and 0.89, with lower values at (19°S, 34°W), respectively 0.64, 0.63 and 0.66. As a whole, the results at (19°S, 34°W) are different from those at the other moorings (Table 3). It could be partly explained by the finding of Foltz et al. (2013) who reported a significant bias at (19°S, 34°W) compared to other moorings despite no apparent dust build-up.

10   The strong correlation is evidenced by the 2D histograms, illustrated in Figure 2 for (6°S, 10°W) for the three data sets, which reveal well aligned distributions with small scattering for both *E* and *KT*. The 2D histograms for the other buoys, and more generally all plots, are quite similar to those of (6°S, 10°W) presented here as examples, except at (19°S, 34°W). All plots are given in Appendix.

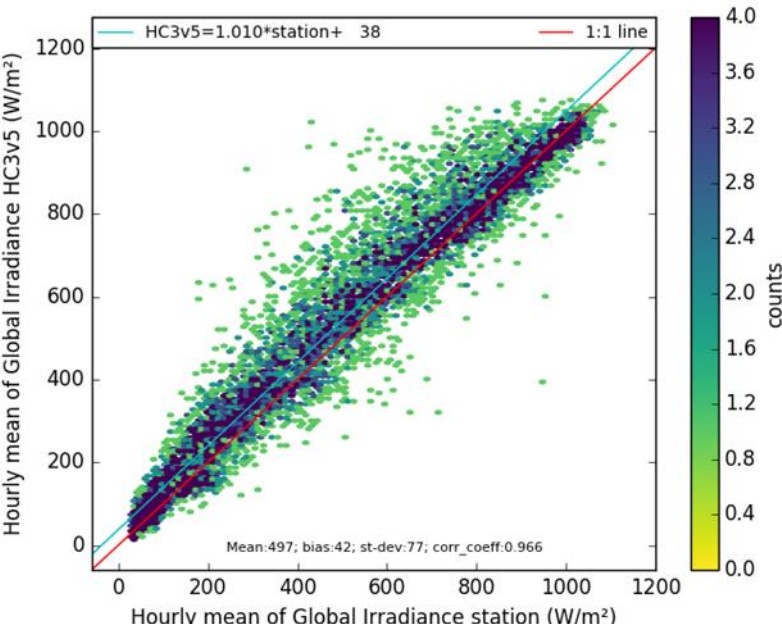

(a)

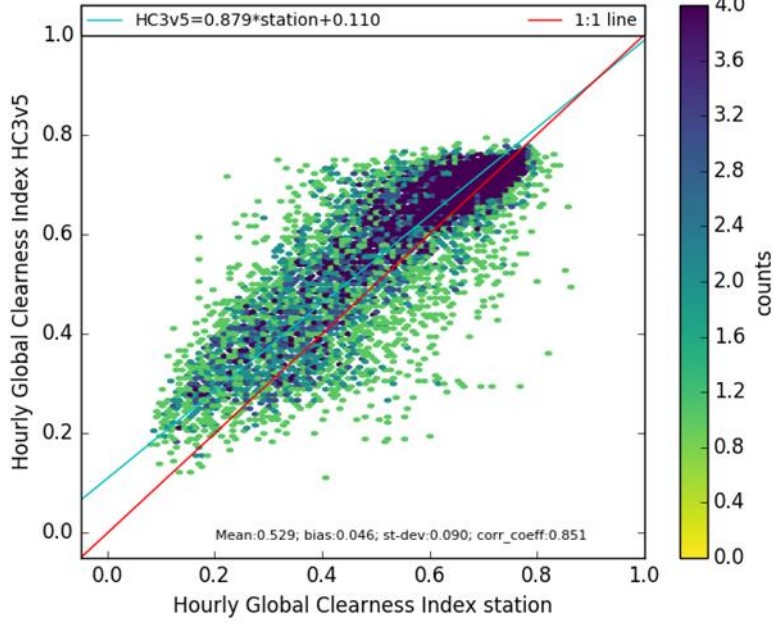

(b)

(c)

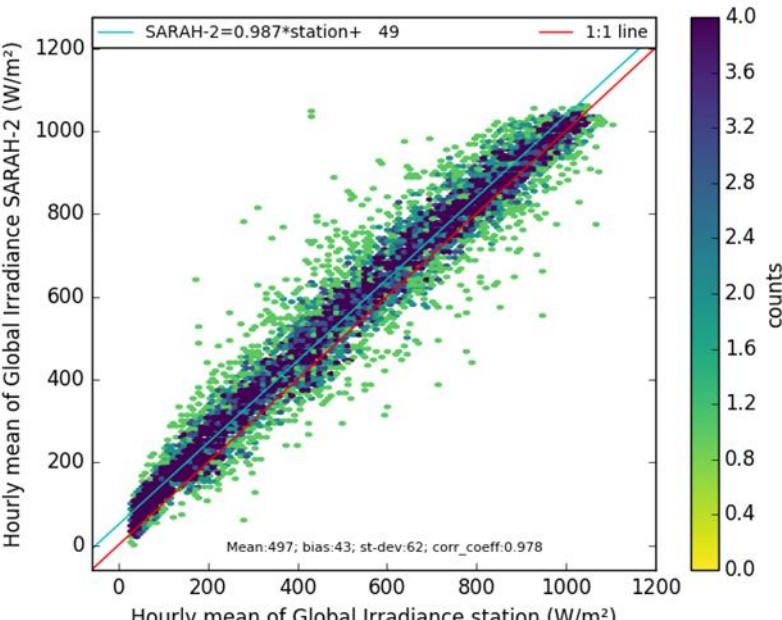

(d)

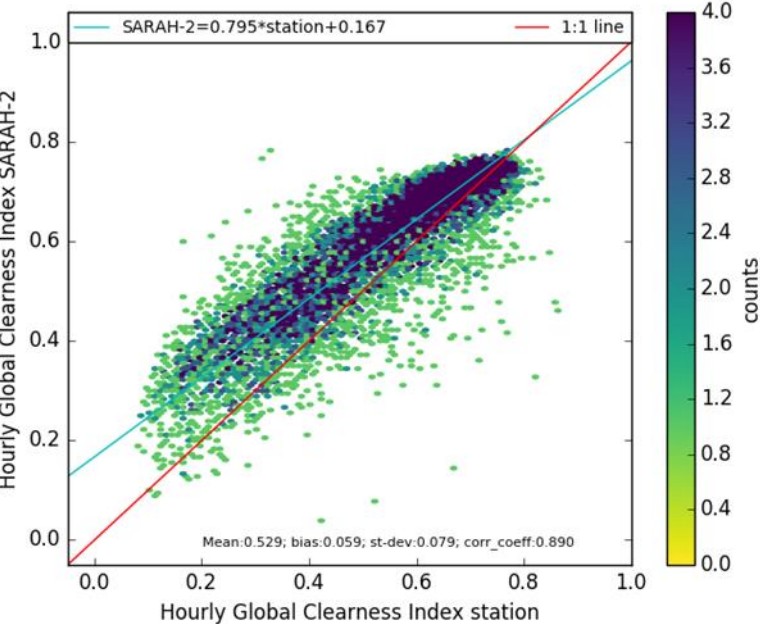

(e)

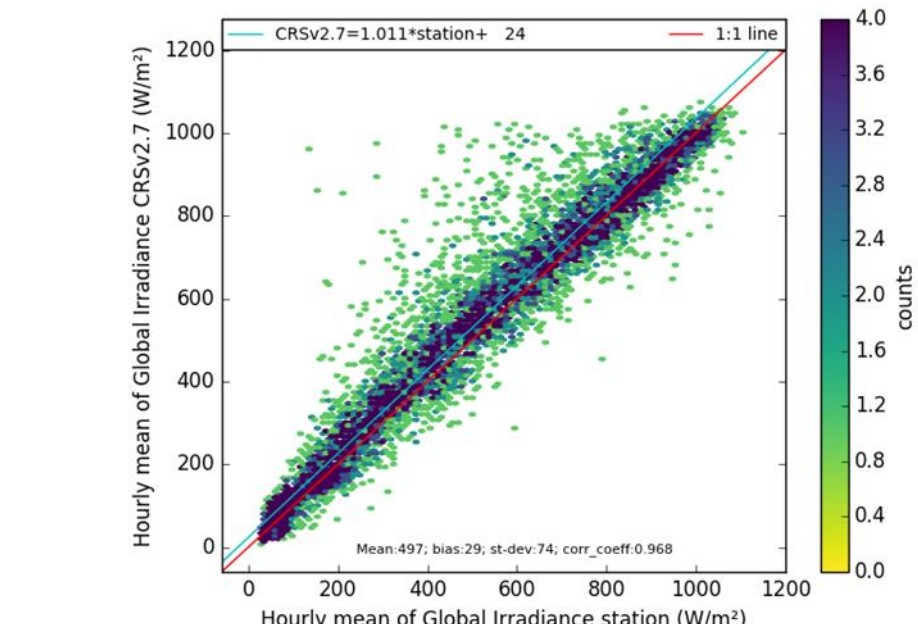

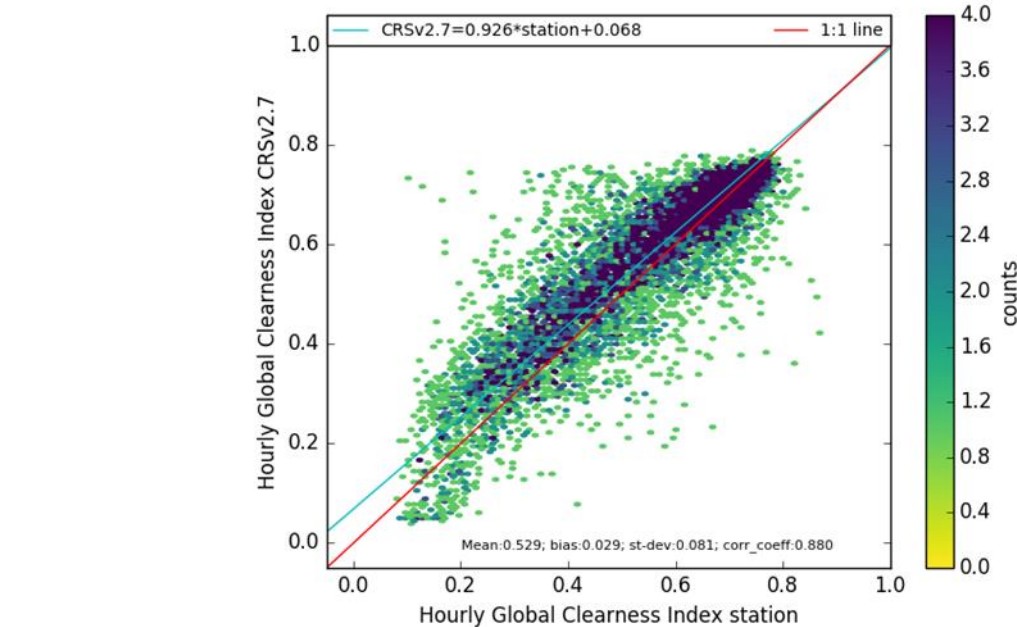

(f)

**Figure 2: 2D histogram of PIRATA measurements (horizontal axis) and data sets (vertical axis) at (6°S, 10°W) for *E* and *KT*. HC3v5: (a) and (b); SARAH-2: (c) and (d); CRS: (e) and (f). Ideally, the dots should lie along the red line (1:1 line). The blue line is the affine function fitted over the points and should ideally overlay the red line.**

One may note an overall overestimation of *E* and *KT* for the three data sets in these graphs and Table 3. The bias in *E* is large and positive. CRS offers the smallest biases with values ranging from 11 to 37 W m⁻²; the bias for SARAH-2 ranges between 12 and 57 W m⁻² and HC3v5 exhibits the greatest biases ranging from 32 to 57 W m⁻².

At a given buoy and a given data set, the errors may depend on the irradiance or clearness index, as shown in Fig. 2 and in

10  Appendix. The irradiances in the range [200, 800] W m$^{-2}$ are overestimated while the greatest irradiances (>800 W m$^{-2}$) are correctly estimated by the three data sets. Both HC3v5 and CRS correctly estimate the smallest irradiances; SARAH-2 tends to overestimate them, except at (19°S, 34°W). The greatest clearness indices are fairly well estimated in the three data sets. There is a tendency to an increasing overestimation with decreasing *KT*. The correct estimation of the greatest *E* and *KT* in HC3v5 and CRS can be related to their use of the McClear clear sky model that estimates the DSIS in cloud-free conditions.

15  This model exploits the properties of the atmosphere delivered by CAMS. Several publications have underlined the good quality of the McClear estimates when compared to high-quality measurements performed at terrestrial stations (Ceamanos et al., 2014; Dev et al., 2017; Eissa et al., 2015a, b; Ineichen, 2016; Lefèvre et al., 2013; Lefèvre and Wald, 2016; Marchand et al., 2017; Zhong and Kleissl, 2015). However, errors are possible in case of any gross errors in aerosol properties provided by CAMS.

The standard deviations of errors in $E$ are fairly similar between the data sets (Table 3). SARAH-2 exhibits the smallest values, ranging from 62 to 79 W m$^{-2}$, with a much greater value at (19°S, 34°W) (119 W m²). HC3v5 and CRS exhibit ranges between 73 and 92 W m$^{-2}$ (117 W m² at (19°S, 34°W)), and 68 and 86 W m$^{-2}$ (113 W m² at (19°S, 34°W)) respectively. The scattering of errors in $KT$ is greater than that in $E$ for the three data sets (Fig. 2). There is no clear relationship between the standard deviation and the frequency of clouds or $KT$ or the geographical location.

This first batch of results deals with pairs of coincident measurements and estimates. In the following, several statistical quantities (frequency distribution, monthly means and standard deviations) are computed on the PIRATA data set and each satellite-derived data set independently and they are compared.

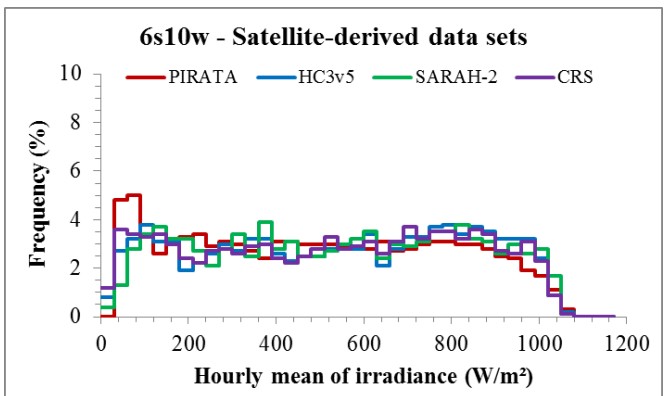 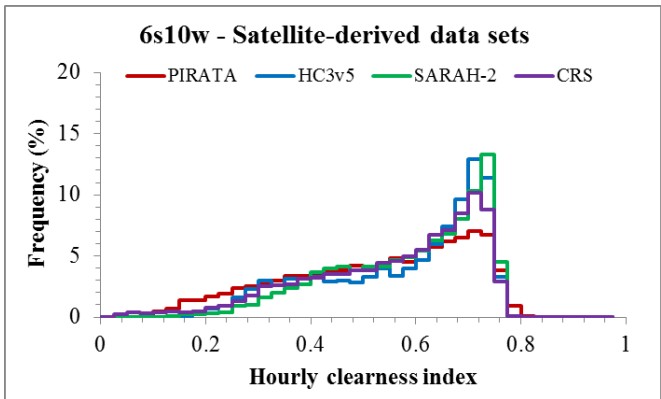

**Figure 3: Frequency distributions of PIRATA measurements (red) and data sets (HC3v5: blue, SARAH-2: green, CRS: purple) at (6°S, 10°W) for *E* (left) and *KT* (right). If the coloured line is above, respectively below, the red one for a given sub-range of values, it means that the data set produces these values too frequently, respectively too rarely with respect to the PIRATA measurements.**

As a whole, the frequency distributions of estimates match well those of the measurements of $E$ for each data set (see Fig. 3 for (6°S, 10°W) and the Appendix). The three data sets exhibit similarities. The frequencies for $E$ less than 100 W m$^{-2}$ are underestimated in the three data sets, except at (0°N, 0°E). As for $KT$, the three data sets exhibit a more or less pronounced underestimation of the frequencies for $KT$ in the range [0.15, 0.4] at (0°N, 0°E), (0°N, 10°W) and (6°S, 10°W). They also exhibit a noticeable overestimation of the frequencies for $KT$ around 0.7, i.e. there are too many cases of cloud-free conditions compared to the PIRATA measurements. Because this happens for the three data sets and at all moorings, it is believed that this is mostly due to the physics in remote sensing and could be explained by the difficulty of accounting for the intra-pixel clouds.

As a whole, the monthly means and standard deviations of $E$ estimated by the three satellite-derived data sets exhibit the same variations throughout the year than those from the PIRATA measurements. The three satellite-derived data sets tend to overestimate the monthly means of $E$ all year long with a few exceptions depending on the mooring and the data set (see Fig. 4 for (6°S, 10°W) and Appendix). As for the standard deviations, SARAH-2 is close to those from PIRATA, with a tendency to overestimation at (0°N, 0°E) and (0°N, 10°W) from December to February. The standard deviations from CRS

are close to those from PIRATA at (0°N, 23°W) and (19°S, 34°W) and at all moorings from March (January at (6°S, 10°W)) to September; there is an overestimation otherwise. As a whole, HC3v5 overestimates the standard deviations from PIRATA measurements, except at (0°N, 23°W) and at the other moorings from March (January at (6°S, 10°W)) to September.

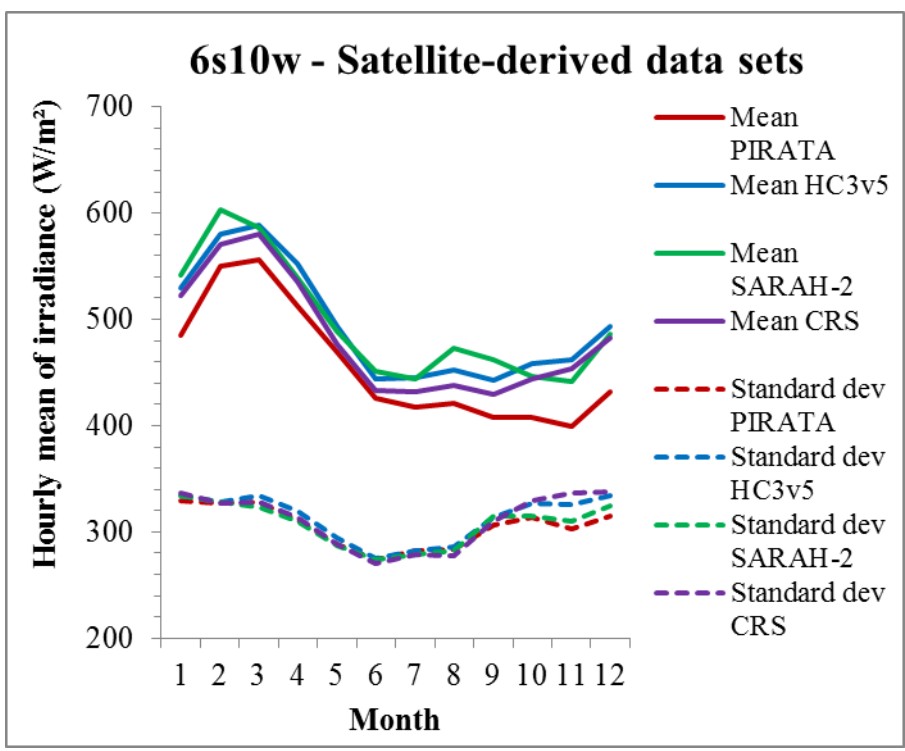

**Figure 4: Monthly means (line) and standard deviations (dotted line) of hourly DSIS, in W m⁻², from PIRATA measurements (red) and data sets (HC3v5: blue; SARAH-2: green; CRS: purple) at (6°S, 10°W). A difference between red line (measurements) and coloured line (data set) for a given month denotes a systematic error for this month: underestimation if the coloured line is below the red line, overestimation otherwise. For a given month, a coloured dotted line above the red one means that the data set produces too much variability for this month; in the opposite case, the data set does not contain enough variability.**

### 4.1.2 Daily values

Table 4 shows the correlation coefficients, the biases and the standard deviations at each buoy for the three satellite-derived data sets, for daily means of *E* and *KT* (correlation coefficient only) in the period 2012-2013. These quantities are similar to

15 those for 2010 and 2011 (not presented).

| Buoy | Correlation coefficient | | | Bias | | | Standard deviation | | |
|---|---|---|---|---|---|---|---|---|---|
| | HC3v5 | SARAH-2 | CRS | HC3v5 | SARAH-2 | CRS | HC3v5 | SARAH-2 | CRS |
| (0°N, 0°E) | 0.916 | 0.922 | 0.911 | 24 | 28 | 16 | 21 | 21 | 21 |
| | (0.914) | (0.920) | (0.910) | (11 %) | (13 %) | (8 %) | (10 %) | (10 %) | (10 %) |
| (0°N, 10°W) | 0.869 | 0.885 | 0.881 | 27 | 28 | 18 | 22 | 21 | 22 |
| | (0.882) | (0.895) | (0.894) | (12 %) | (12 %) | (8 %) | (10 %) | (9 %) | (9 %) |
| (0°N, 23°W) | 0.875 | 0.921 | 0.903 | 15 | 13 | 9 | 17 | 14 | 15 |
| | (0.884) | (0.926) | (0.910) | (6 %) | (5 %) | (3 %) | (7 %) | (5 %) | (6 %) |
| (6°S, 10°W) | 0.914 | 0.939 | 0.920 | 20 | 21 | 14 | 18 | 16 | 18 |
| | (0.911) | (0.935) | (0.922) | (9 %) | (9 %) | (6 %) | (8 %) | (7 %) | (8 %) |
| (19°S, 34°W) | 0.910 | 0.905 | 0.915 | 16 | 6 | 6 | 27 | 28 | 27 |
| | (0.796) | (0.789) | (0.808) | (7 %) | (3 %) | (2 %) | (11 %) | (12 %) | (11 %) |

**Table 4: For each PIRATA buoy and each satellite-derived data set: Correlation coefficient between measurements and estimates from satellite-derived data sets for irradiance and clearness index *(in italics),* bias and standard deviation ($W\,m^{-2}$) between measurements and estimates for irradiance with relative values (in brackets) for daily means in the period 2012-2013.**

As expected, one may note that for a given data set and a given PIRATA mooring, the numbers in this Table are consistent between hourly and daily values: the relative biases are the same for the hourly and daily means, and the standard deviations of errors are less for daily values than for hourly values because of the averaging over the day (Tables 3, 4). There is also an agreement between the estimations of the monthly means and standard deviations of $E$ from hourly and daily values (not shown).The three estimates correlate very well with the measurements and offer very similar correlation coefficients for both

$E$ and $KT$. The correlation coefficients for the daily $E$ range between 0.87 and 0.92 for HC3v5, 0.89 and 0.94 for SARAH-2, and 0.89 and 0.92 for CRS (Table 4). The correlation coefficients for daily $E$ are less than those for hourly $E$ because of the strong influence of the solar zenithal angle which creates a de facto correlation between estimates and measurements in case of hourly values. The correlation coefficients for daily $KT$ are greater than for hourly $KT$, close to or greater than 0.9 for the three satellite-derived data sets (0.8 at (19°S, 34°W)), and similar to those for the daily $E$ (Table 4).

The 2D histograms show that the three data sets overestimate the daily $E$ and $KT$ (not shown). The frequency distributions of daily $E$ and $KT$ of the three data sets are not very close to those from PIRATA as a whole. There is a lack of frequencies for $E$<250 $W\,m^{-2}$ and for $KT$<0.6, and too many frequencies for $E$>250 $W\,m^{-2}$ and for $KT$>0.6.

### 4.1.3 Discussions on the three satellite-derived data sets

Thomas et al. (2016) have performed comparisons of hourly and daily measurements of $E$ performed at a total of 42
Brazilian stations, i.e. at similar latitudes, against estimates from HC3v5 on the one hand, and CRS on the other hand. The reported performances are fairly similar to those of the present work for both hourly and daily $E$. One may note that the

biases at terrestrial sites are closer to 0 than those for the PIRATA buoys and that the standard deviations are a bit smaller. This may indicate some limitations in the accuracy of the PIRATA measurements.

Though care should be taken given the small number of buoys, one may note a tendency for the bias to decrease from East to West. This could be related to the increase in the mean *KT* from East to West as shown in Table 1. The tendency is more pronounced for SARAH-2 than for HC3v5 and CRS. For CRS, this tendency is in agreement with the CAMS validation results reported quarterly since 2014 by Lefèvre and Wald using terrestrial stations (https://atmosphere.copernicus.eu/validation-supplementary-products). Though it has not been discussed by these two authors, one may note that their reports indicate a tendency of the bias to decrease with an increase in the mean *KT*. The tendency is more visible at the terrestrial stations experiencing frequent cloud-free conditions similarly to the selected PIRATA buoys.

The differences between the greatest and smallest biases are 25 W m$^{-2}$ for HC3v5, 45 W m$^{-2}$ for SARAH-2, and 26 W m$^{-2}$ for CRS respectively. If one removes the worst cases, the differences amount to 16, 29 and 19 W m$^{-2}$ respectively. The small differences in bias over the sites for HC3v5 and CRS means that the bias is almost constant in space and that the spatial features of *E* are fairly well reproduced by HC3v5 and CRS as a whole as the spatial variability, expressed e.g. by spatial gradients, is not artificially distorted by artefacts due to spatial variations in bias.

The large correlation coefficients mean that the time-series of the actual field of DSIS are well reproduced by any of the three data sets at hourly and daily timescales though amplitudes of variation in time may be hampered by the large standard deviation of the errors (Tables 3 and 4). This finding is consistent with those of Bengulescu et al. (2017) who reported very high correlation coefficients between HC3v5 and in situ measurements at various timescales, from days to years. A similar conclusion may be drawn for the clearness index.

The frequency distributions of *E* and *KT* are fairly well reproduced in each satellite-derived data set though one may note a noticeable overestimation of the number of cloud-free cases compared to the PIRATA measurements likely due to the difficulty of accounting for the intra-pixel clouds. The monthly standard deviations of *E* are also well reproduced while the situation on the monthly means is more mixed depending on the data set.

## 4.2.    The re-analysis data sets (MERRA-2, ERA5)

### 4.2.1. Hourly values

Table 5 shows the correlation coefficients, the biases and the standard deviations at each buoy for the two re-analysis data sets, for hourly means of *E* and *KT* (correlation coefficient only) in the period 2012-2013. These quantities are similar to those for 2010 and 2011 (not presented).

| Buoy | Correlation coefficient | | Bias | | Standard deviation | |
|---|---|---|---|---|---|---|
| | MERRA-2 | ERA5 | MERRA-2 | ERA5 | MERRA-2 | ERA5 |
| (0°N, 0°E) | 0.826 *(0.475)* | 0.879 *(0.597)* | -44 (-10 %) | 23 (5 %) | 163 (36 %) | 138 (30 %) |
| (0°N, 10°W) | 0.853 *(0.480)* | 0.902 *(0.627)* | -18 (-4 %) | 24 (5 %) | 151 (31 %) | 125 (25 %) |
| (0°N, 23°W) | 0.898 *(0.529)* | 0.923 *(0.621)* | -8 (-2 %) | -6 (-1 %) | 129 (24 %) | 113 (21 %) |
| (6°S, 10°W) | 0.887 *(0.542)* | 0.889 *(0.562)* | 13 (3 %) | 16 (3 %) | 136 (27 %) | 134 (27 %) |
| (19°S, 34°W) | 0.908 *(0.499)* | 0.925 *(0.633)* | 23 (4 %) | -10 (-2 %) | 128 (25 %) | 115 (23 %) |

**Table 5: For each PIRATA buoy and each re-analysis data set: Correlation coefficient between measurements and estimates from re-analysis for irradiance and clearness index *(in italics)*, bias and standard deviation (W m$^{-2}$) between measurements and estimates for irradiance with relative values (in brackets) for hourly means in the period 2012-2013.**

5 The two estimates correlate well with the PIRATA measurements for *E* with correlation coefficients ranging between 0.83 and 0.91 for MERRA-2, and between 0.88 and 0.93 for ERA5. The correlation coefficients are weaker for *KT*: from 0.48 to 0.54 for MERRA-2, and from 0.56 to 0.63 for ERA5. It means that at most 29 % and 40 % of the variance contained in the measured clearness indices is explained by MERRA-2 and ERA5 respectively. One may note that the correlation coefficients for MERRA-2 exhibit a tendency to increase from East to West, both in *E* and *KT*.

10 The strong correlation in *E* and the weak correlation in *KT* are evidenced in the 2D histograms, illustrated in Figure 5 for (6°S, 10°W) for the two data sets. The 2D histograms at the other buoys, and more generally all plots, are quite similar to those of (6°S, 10°W) presented here as examples, except at (19°S, 34°W). All plots are given in Appendix.

The 2D histograms show that the dots for *E* are fairly well aligned along the 1:1 line with a very large scattering (see e.g. Fig. 5a, c). One may note a large underestimation of the greatest DSIS, i.e. greater than 700-800 W m$^{-2}$ for both data sets. As

15 for *KT*, the 2D histograms for MERRA-2 show shapes that are not well elongated and are more like rectangles or triangles along lines whose slopes are less than 1 (Fig. 5b). The 2D histograms for ERA5 are more elongated along lines whose slopes are less than 1 (Fig. 5d) but still exhibit very large scattering.

(a)

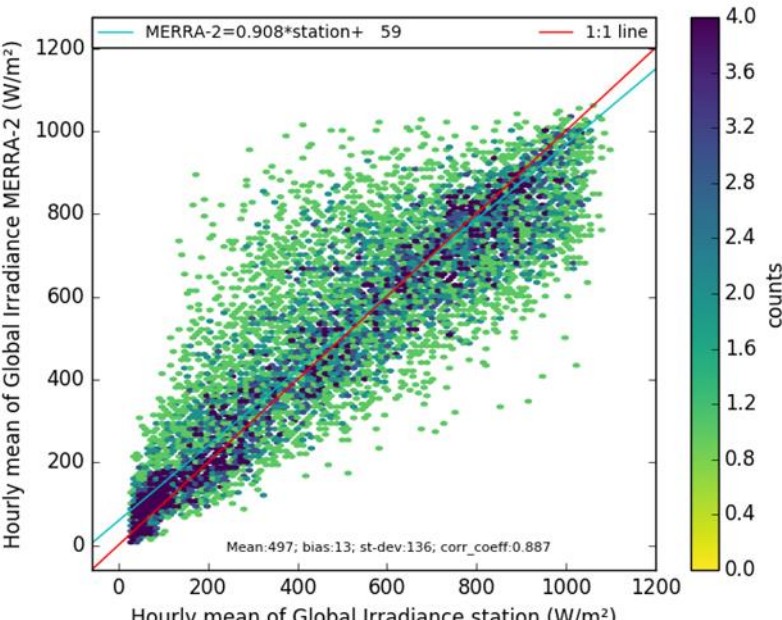

(b)

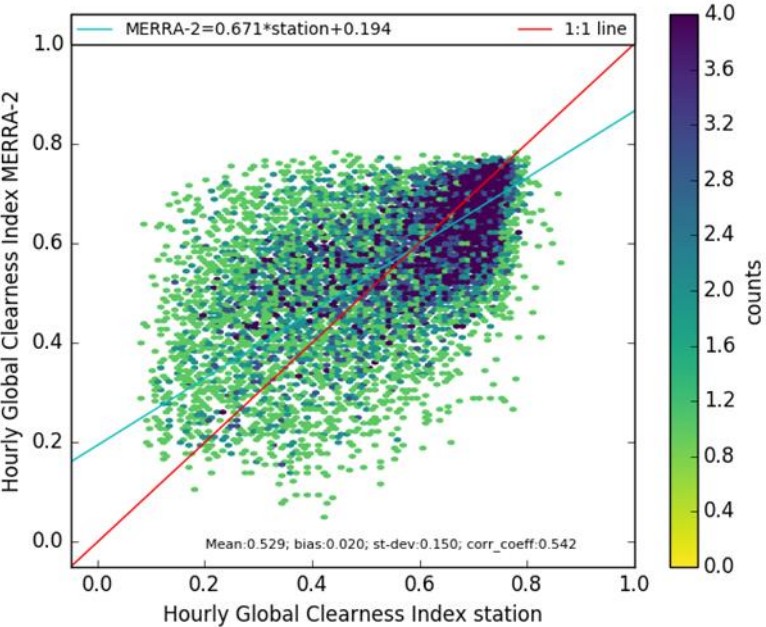

(c)

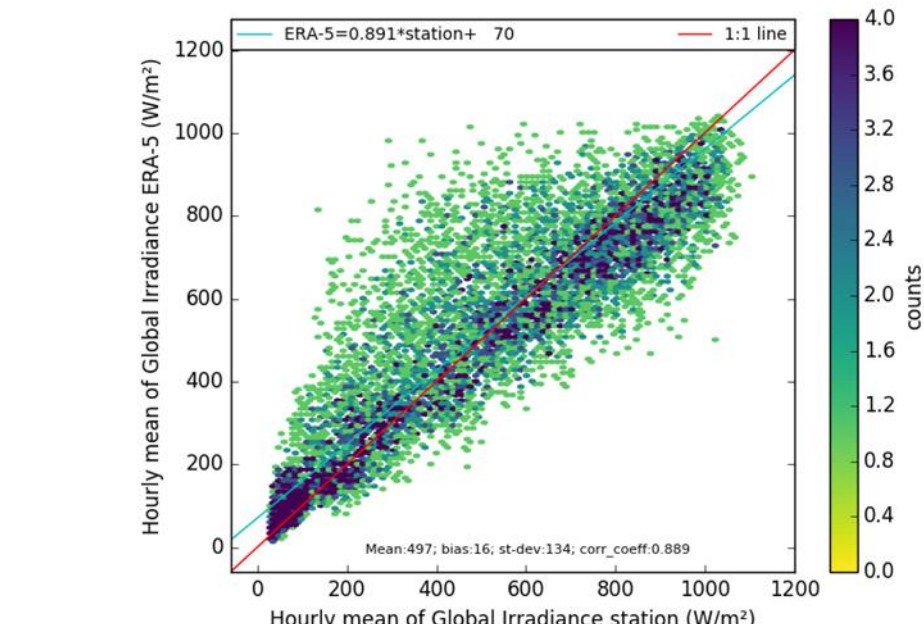

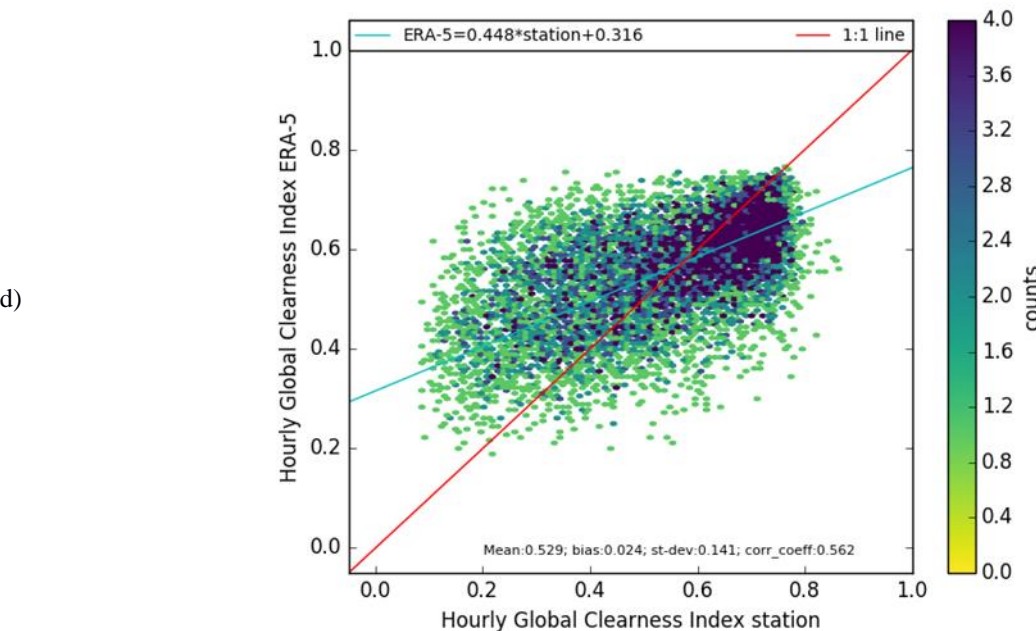

(d)

**Figure 5: 2D histogram of PIRATA measurements (horizontal axis) and data sets (vertical axis) at (6°S, 10°W) for *E* and *KT*. MERRA-2: (a) and (b); ERA5: (c) and (d). Ideally, the dots should lie along the red line (1:1 line). The blue line is the affine function fitted over the points and should ideally overlay the red line.**

One may ask if the underestimation of the greatest *E* originates from the model used in the re-analysis for cloud-free conditions. As MERRA-2 provides the hourly DSIS in cloud-free conditions, a comparison was made with the hourly DSIS from McClear. Both data sets reveal similar irradiances. It is concluded that the MERRA-2 cloud-free DSIS are likely accurate and that the underestimation of the greatest *E* is mostly due to errors in the assessment of cloud properties by MERRA-2. This is in agreement with the analysis of the 2D histograms for *KT* which exhibit a well-marked underestimation of the greatest *KT*.

ERA5 does not provide estimates of the DSIS for cloud-free conditions in contrary to MERRA-2. Hence, individual days have been selected randomly for which the daily profiles were similar to the expectations for cloud-free conditions based on a visual analysis. The daily profiles of the DSIS from McClear were superimposed. It was found that the DSIS for cloud-free conditions is underestimated by ERA5. This is supported by the analysis of the 2D histograms for *KT* (see e.g. Fig. 5d) which shows a tendency to overestimate for *KT*<0.6 and a well-marked underestimation for *KT*>0.6. These observations are clearly seen on the frequency distribution to be discussed later (Fig. 7).

The bias in *E* varies strongly with the mooring and with the data set. It ranges between -44 and 23 W m⁻² for MERRA-2 and between -10 and 23 W m⁻² for ERA5. The 2D histograms show a tendency for both data sets to overestimate the smallest irradiances and to underestimate the greatest ones. Both re-analyses tend to overestimate *KT* but the situation is complex,

especially in MERRA-2 where the error looks a bit random, i.e. there is as much chances of observing an overestimation of the actual clearness index as an underestimation.

The standard deviations of the errors for ERA5 are comprised between 113 (21 %) and 138 W m² (30 %). Those for MERRA-2 are greater: they range from 128 (25 %) to 163 W m² (36 %). Further exploration shows the dependency of the

errors for MERRA-2 to the differences between the true solar time and the mean solar time. MERRA-2 does not account for this difference which is a function of the day in the year as a first approximation and is ranging from -17 to 17 min (Fig. 6). The true solar time is that needed for computing the solar zenithal angle accurately enough while it seems that MERRA-2 performs only a rough estimate of this angle by using the mean solar time. This angle intervenes twice: firstly to compute the irradiance impinging on the horizontal plane at the top of atmosphere and secondly as a major input to the radiative transfer

model. Hence, an error in this angle yields an error in the estimated DSIS. This weakens the correlation between the PIRATA measurements and the MERRA-2 data set and it increases the standard deviation of errors for both hourly and daily values of the DSIS.

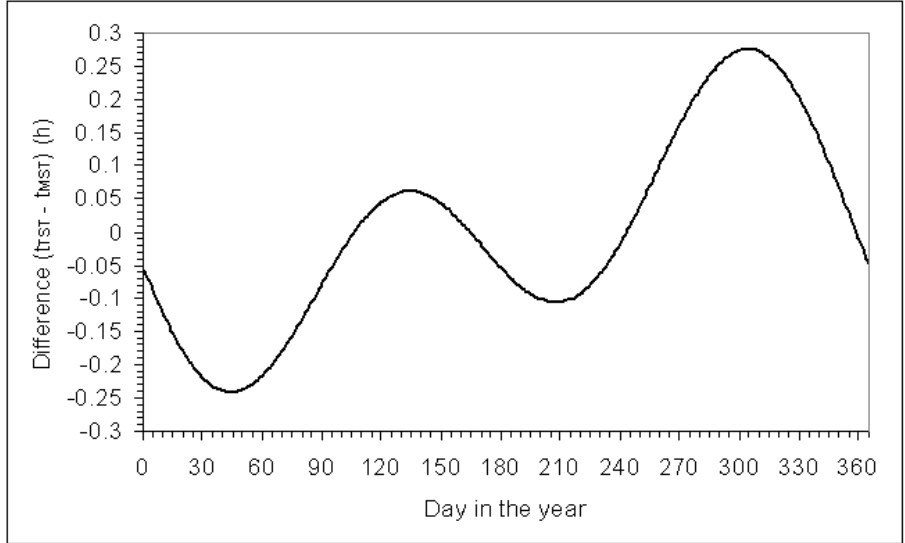

**Figure 6: Difference between the true solar time ($t_{TST}$) and the mean solar time ($t_{MST}$). Excerpt from Wald (2007).**

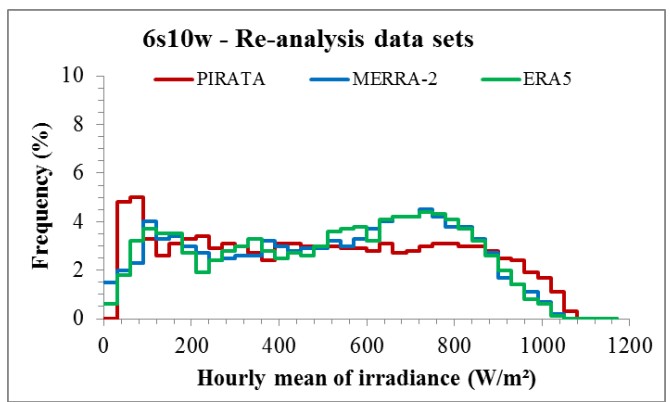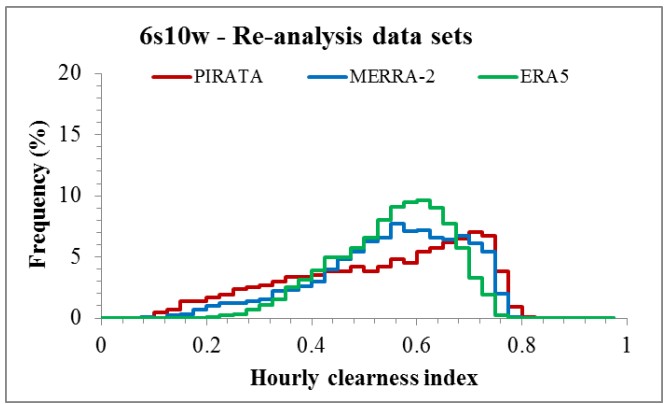

**Figure 7: Frequency distributions of PIRATA measurements (red) and data sets (MERRA-2: blue, ERA5: green) at (6°S, 10°W) for *E* (left) and *KT* (right). If the coloured line is above, respectively below, the red one for a given sub-range of values, it means that the data set produces these values too frequently, respectively too rarely with respect to the PIRATA measurements.**

The agreement between the frequency distributions of *E* and *KT* from measurements and those from the data sets depend on the mooring and the data set (see Fig. 7 for (6°S, 10°W) and Appendix). The agreement for *E* is fairly good for MERRA-2 at (0°N, 10°W), (0°N, 23°W), and (19°S, 34°W). Otherwise, there are too many frequencies for *E*<250 W m$^{-2}$ and too many missing for *E*>250 W m$^{-2}$ at (0°N, 0°E) while it is the opposite at (6°S, 10°W). At (0°N, 0°E), (0°N, 10°W) and (0°N, 23°W), there are too many frequencies for *KT* in the range [0.50, 0.65] and not enough when *KT*>0.65, i.e. there are not

enough cases of cloud-free conditions. On the contrary, MERRA-2 underestimate the frequencies for *KT*<0.45 -not enough cases of overcast conditions- and overestimate the frequencies in the range [0.45, 0.7] at 6s10e and (19°S, 34°W). For ERA5, frequencies are missing for *E*<200 W m$^{-2}$ and there are too many frequencies around 250 W m$^{-2}$. The estimates by ERA5 underestimate the frequencies for *KT*<0.4 and *KT*>0.7, and overestimate the frequencies for *KT* in-between.

The two re-analyses differ between them regarding the estimation of the monthly means and standard deviations of *E* and

each data set does not exhibit the same features at all moorings (see Fig. 8 for (6°S, 10°W) and Appendix). The monthly means and standard deviations estimated by MERRA-2 exhibit the same variations than those from PIRATA at (0°N, 23°W), (6°S, 10°W), and (19°S, 34°W). At (0°N, 0°E) and (0°N, 10°W), there is an anti-correlation for both the means and the standard deviations between January and May. There is no general pattern for MERRA-2; it depends on the mooring. As a whole, one may note a tendency to underestimation for the first months and to overestimation or agreement at the end of

the year for both the means and the standard deviations.

At (0°N, 0°E), the estimated means underestimate those from PIRATA all year long, except in June, November and December where there is an agreement. At (0°N, 10°W), the underestimation is from February to July, then follows a slight overestimation from August to October, and a more noticeable overestimation from November to January.

As a whole, the monthly means and standard deviations of *E* estimated by ERA5 exhibit the same variations throughout the

25  year than those from the PIRATA measurements at all moorings. The standard deviations are most often underestimated by

ERA5. The situation is more contrasted for the means with a tendency to overestimation at (0°N, 0°E), (0°N, 10°W), and (6°S, 10°W), and an agreement or underestimation at (0°N, 23°W) and (19°S, 34°W).

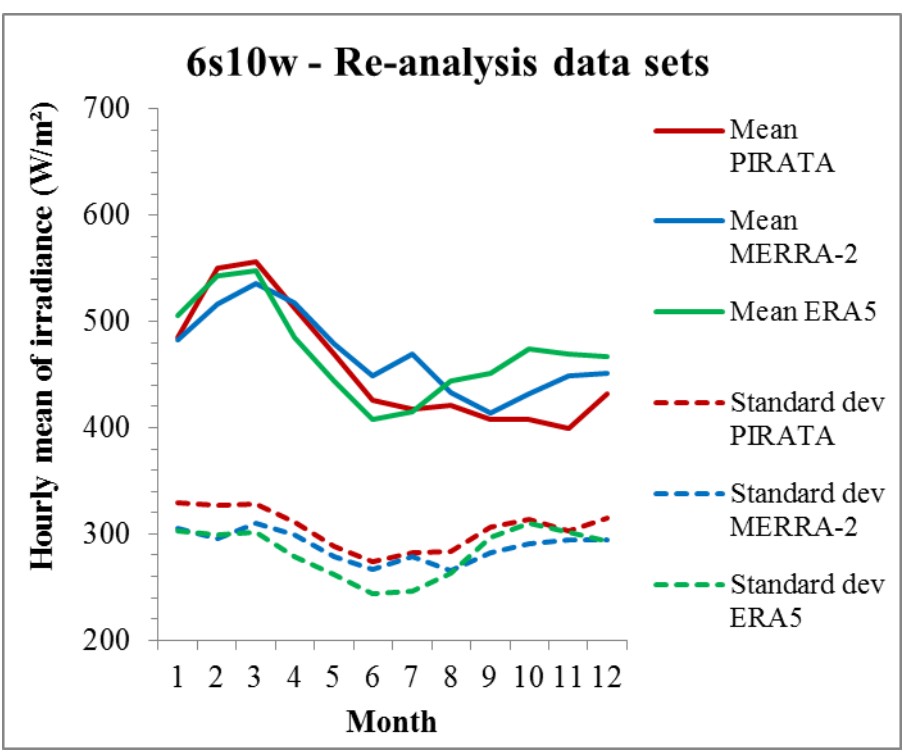

**Figure 8: Monthly means (line) and standard deviations (dotted line) of hourly DSIS, in W m$^{-2}$, from PIRATA measurements (red) and data sets (MERRA-2: blue, ERA5: green) at (6°S, 10°W). A difference between red line (measurements) and coloured line (data set) for a given month denotes a systematic error for this month: underestimation if the coloured line is below the red line, overestimation otherwise. For a given month, a coloured dotted line above the red one means that the data set produces too much**
10 **variability for this month; in the opposite case, the data set does not contain enough variability.**

### 4.2.2. Daily values

Table 6 shows the correlation coefficients, the biases and the standard deviations at each buoy for the two re-analyses data sets, for daily means of $E$ and $KT$ (correlation coefficient only) in the period 2012-2013. These quantities are similar to those

15 for 2010 and 2011 (not presented).

| Buoy | Correlation coefficient | | Bias | | Standard deviation | |
|---|---|---|---|---|---|---|
| | MERRA-2 | ERA5 | MERRA-2 | ERA5 | MERRA-2 | ERA5 |
| (0°N, 0°E) | 0.315 *(0.340)* | 0.587 *(0.586)* | -17 (-8 %) | 14 (7 %) | 57 (27 %) | 43 (20 %) |
| (0°N, 10°W) | 0.290 *(0.350)* | 0.490 *(0.545)* | -7 (-3 %) | 13 (6 %) | 53 (23 %) | 40 (17 %) |
| (0°N, 23°W) | 0.353 *(0.392)* | 0.504 *(0.542)* | -3 (-1 %) | -2 (-1 %) | 41 (16 %) | 31 (12 %) |
| (6°S, 10°W) | 0.477 *(0.462)* | 0.475 *(0.368)* | 7 (3 %) | 8 (3 %) | 42 (18 %) | 41 (17 %) |
| (19°S, 34°W) | 0.799 *(0.575)* | 0.864 *(0.691)* | 12 (5 %) | -4 (-2 %) | 40 (17 %) | 33 (14 %) |

**Table 6: For each PIRATA buoy and each re-analysis data set: Correlation coefficient between measurements and estimates from re-analysis for irradiance and clearness index *(in italics),* bias and standard deviation (W m$^{-2}$) between measurements and estimates for irradiance with relative values (in brackets) for daily means in the period 2012-2013.**

As expected and similarly to the case of the satellite-derived data sets, for a given data set and a given PIRATA mooring, the numbers in this Table are consistent between hourly and daily values: the relative biases are the same for the hourly and daily means, and the standard deviations of errors are less for daily values than for hourly values (Table 5, 6). There is also an agreement between the estimations of the monthly means and standard deviations of $E$ from hourly and daily values (not shown).

The correlation is weak between each re-analysis data set and the PIRATA measurements: the correlation coefficients for the daily DSIS range from 0.29 to 0.48 (0.80 at (19°S, 34°W)) for MERRA-2, and from 0.48 to 0.59 (0.86 at (19°S, 34°W)) for ERA5. They are less than those for the hourly DSIS. The correlation coefficients are also low for the daily $KT$: from 0.34 to 0.46 (0.58 at (19°S, 34°W)) for MERRA-2 and from 0.37 to 0.59 (0.69 at (19°S, 34°W)) for ERA5. At most 23 %, respectively 21 %, of the variance contained in the measured daily $E$ and $KT$ are explained by the estimates from MERRA-2 (74 % and 34 % at (19°S, 34°W)), which is very little. It is more for ERA5 with 35 % for both $E$ and $KT$ (74 % and 48 % at (19°S, 34°W)) but is still low. There is no regional trend of the correlation coefficients for any of the two data sets. The 2D histograms (not shown) have shapes which are not elongated and look more like discs for both $E$ and $KT$. They exhibit very large scattering, in full agreement with the low correlation coefficients. For both data sets, $E$, respectively $KT$, is sometimes overestimated but more frequently underestimated, especially for the greatest values.

### 4.2.3. Discussion on the two re-analyses data sets

One striking feature observed in both re-analyses data sets is that cloud-free conditions are often reported by these data sets while the actual conditions are cloudy and reciprocally, actual cloud-free conditions are reported as cloudy, at both hourly and daily timescales. Similar observations were reported for daily DSIS over the Atlantic Ocean by Boilley and Wald (2015) for the MERRA and ERA-Interim re-analyses.

The bias in hourly $E$ ranges between -44 and 23 W m$^{-2}$ for MERRA-2 and shows a tendency to increase (from negative to positive) from East to West. The dependency of the bias with the solar zenithal angle and other variables is weak, except for

the cloud type whose influence is prominent. Fig. 9 shows the dependency of the bias (left) and of the correlation coefficient (right) as a function of the cloud type from the CAMS cloud classification at each mooring for MERRA-2. A column would be uniformly coloured in case of no dependency of the bias (left) or correlation coefficient (right) with the type of cloud. A row would be uniformly coloured in case of no dependency with the mooring.

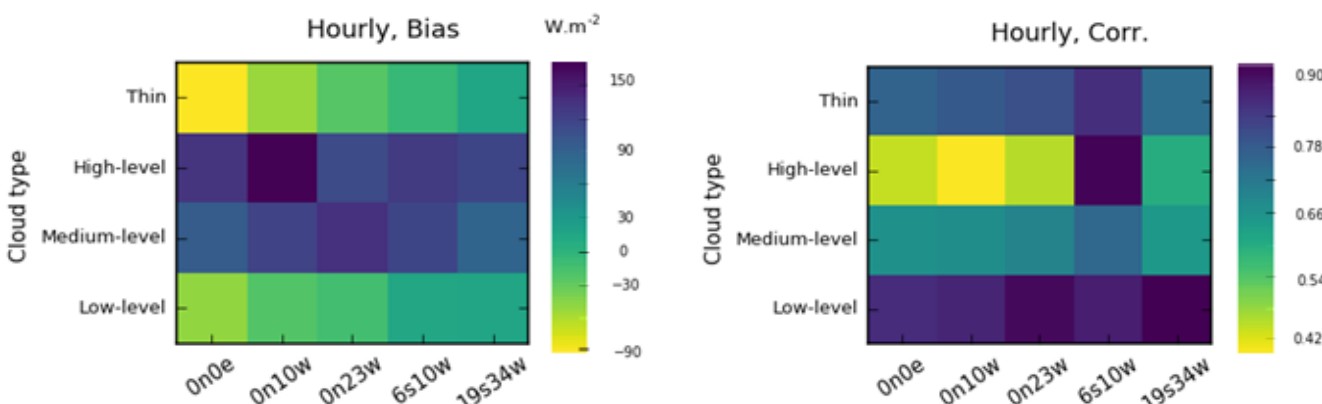

**Figure 9: Bias (left, in W m$^{-2}$) and correlation coefficient (right) as a function of the cloud type at each PIRATA buoy for MERRA-2.**

One observes greenish tones in the lowest row in Fig. 9 (left) for the 'low-level' type (water cloud at low altitude). This
means biases close to 0 for this type at each mooring. There is a slight change in the green tones over the row, meaning that the bias does not vary much from one mooring to another for this type of cloud though the tendency to increase (from negative to positive) from East to West (from left to right) is noticeable. The tones are blueish for the 'medium-level' (water cloud at medium altitude) and 'high-level' (deep cloud of large vertical extent from low altitude to medium altitude), meaning large positive biases, with changes from site to site without any clear trend. The 'thin' type (thin ice cloud) exhibits
yellow to greenish tones, with negative bias. One may note the increase of the bias (from negative to positive) from East to West.

Fig. 9 (right) shows that the correlation coefficients are similar or very close for all moorings (each row is fairly uniformly coloured), except for 'high-level' clouds for which the coefficients vary very much with the mooring without a clear trend. This type of cloud exhibits the lowest correlation coefficients while the 'low-level' and 'thin' types offer the greatest ones.
The fact that the 'medium-level' and 'high-level' clouds exhibit more bias and less correlation than the 'low-level' clouds is consistent with the recent and preliminary findings of Doddy et al. (2017) who looked at the differences between measurements of daily $E$ performed at terrestrial stations in Ireland and MERRA-2 outputs and suggested a systematic link between prevailing cloud structures and errors.

The bias in hourly $E$ ranges between -10 and 24 W m⁻² for ERA5. It exhibits a regional tendency to decrease in absolute values and to tend to underestimation with increasing mean $KT$ (Table 5). However, such complex behaviours can only be speculated given the small number of moorings. The dependency of the bias with the solar zenithal angle and other variables is weak, except for the cloud type which is discussed now. Fig. 10 shows the dependency of the bias (left) and of the correlation coefficient (right) as a function of the cloud type from the CAMS cloud classification at each mooring for ERA5.

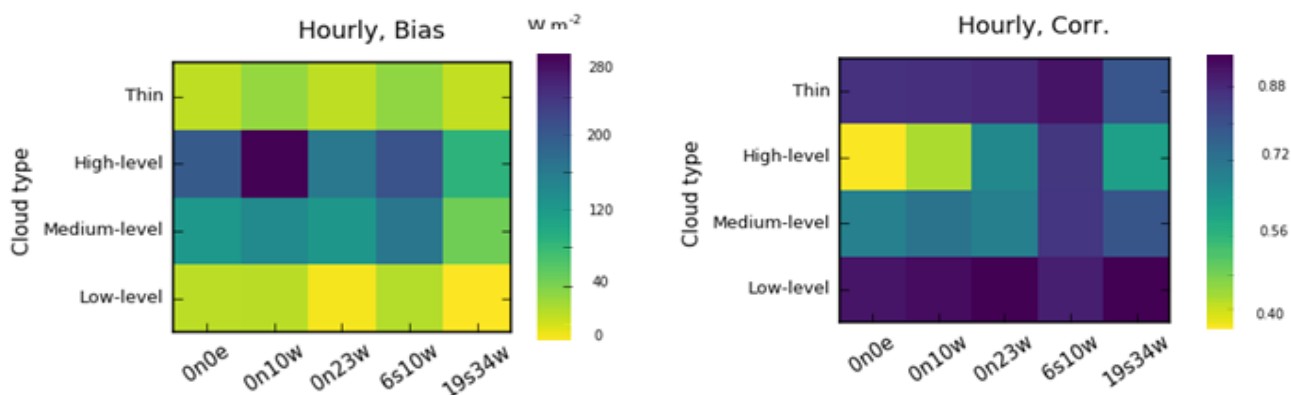

**Figure 10: Bias (left, in W m⁻²) and correlation coefficient (right) as a function of the cloud type at each PIRATA buoy for ERA5.**

One observes yellowish and greenish tones in the lowest row in Fig. 10 (left) for the 'low-level' type for ERA5. The bias is positive and small. The minima are reached at (0°N, 23°W) and (19°S, 34°W) which are the moorings with the greatest mean $KT$ (Table 1). The bias is also small and positive for the 'thin' clouds with a weak dependency upon the moorings. Biases are greater for the two other types and exhibit greater changes from mooring to mooring, though the moorings (0°N, 23°W) and (19°S, 34°W) show lower biases than the others. One may note that the biases in Fig. 10 (left) are positive for every row, i.e. every type, while the bias for all conditions is either negative or positive (Table 5). This is explained by the fact that these moorings exhibit a large number of cloud-free conditions during which the DSIS tends to be underestimated by ERA5 as discussed above. Similarly to MERRA-2, the correlation coefficients (Fig. 10 right) show large values that are fairly the same between moorings for the 'low-level' and 'thin' types, and the correlation is weaker for the two other types with coefficients varying from mooring to mooring without a clear trend.

MERRA-2 exhibits large changes in biases from one mooring to another with a trend from East to West as previously reported; the amplitude of change in bias between moorings is 67 W m⁻² for the hourly DSIS. The high variability of bias in space indicates an artificial spatial distortion of the field of DSIS. Koster (2015) presented an initial evaluation of the climate in MERRA-2. As for radiation, he used the CERES (Clouds and the Earth's Radiant Energy System) EBAF (Energy Balanced and Filled) satellite-based observational data set as a reference. Significant regional biases between the yearly means of the DSIS of MERRA-2 and those from the CERES data set were reported by Koster. In his Figure 4.6, one may see

a noticeable difference between both data sets. It ranges from -20 W m⁻² in the Gulf of Guinea to 20 W m⁻² along the Brazilian coast. The findings reported in the present work give some flesh to the work of Koster as the magnitude of this difference and its spatial structure are supported by the present work which is built on in situ measurements.

The spatial distortion in MERRA-2 combines with very large standard deviations of errors, about 140 W m$^{-2}$, and this may indicate that the spatial distortion occurs at various timescales. This is supported by the findings of Bengulescu et al. (2017). These authors performed a comparison between several data sets, among which HC3v5 and MERRA-2, and in situ measurements made at two terrestrial stations: Vienna (Austria) and Kishinev (Moldova). They reported a very high correlation coefficient between MERRA-2 and in situ measurements (0.97 and 0.97 respectively) and showed that this high correlation was mostly due to the very high correlation coefficients between MERRA-2 and in situ measurements at the yearly period (0.99 at both stations), i.e. MERRA-2 reproduced well the seasonal variability. For any timescale less than 1 year, the correlation coefficient reported by these authors was less than 0.8, which means that less than 64 % of the variance of the measurements was explained by MERRA-2, i.e. MERRA-2 did not reproduce the variability observed in measurements for timescales shorter than 1 year.

The difference in bias for the hourly DSIS from one mooring to another is 34 W m$^{-2}$ for ERA5. This change in bias in space means that the spatial features of $E$ are partly artificially distorted. Though the correlation coefficients are close to 0.90 for hourly $E$, the large scattering of errors (standard deviation of errors around 125 W m$^{-2}$ for hourly $E$) combined with the changes in bias and the low correlation for the daily DSIS hampers the use of ERA5 in detailed studies of the spatial and temporal variability of the DSIS.

## 4.3. About the differences in spatial support of the buoy and the grid cell of the data sets

One may object that the size of the grid cell is inappropriate for the comparison with a single buoy because surface measurements are for a single point in space, whereas the estimated irradiances are for the area of a pixel (typically 5 km) or a grid cell (typically 50 km). Cloud properties may vary within the grid cell and large random errors are unavoidable at hourly time steps. Using monthly averages is a means to reduce the errors caused by the problem (see e.g. Zhao et al., 2013). One may believe that this mismatch in spatial support of information may explain the performances of the re-analyses presented here. However, it can be argued that there is no orographic effect in the Atlantic Ocean and there is no strong systematic gradient in irradiance over short distances corresponding to the hourly time step. Hence, the irradiance field is fairly homogeneous at sub-meso-scales and this should mitigate the effects of the differences in spatial support of the buoy and the grid cell. In addition, one may note that the drawbacks reported above are also observed at daily timescale. Finally, the work of Boilley and Wald (2015) can be mentioned. These authors compared the satellite-derived HelioClim-1 data set to PIRATA measurements. HelioClim-1 is fairly similar to the re-analyses with regard to the spatial support of information because it is made of estimates of the DSIS made on 5 km pixels spaced by 25 km in both latitude and longitude (Lefèvre et al., 2007, 2014), and a spatial bi-linear interpolation was performed to create the time-series at the locations of the PIRATA

moorings. Though the period is not the same as in the present study (HelioClim-1 covers the period 1985-2005), one may compare the correlation coefficients reported by these authors that range between 0.82 and 0.88 for daily *E* and from 0.79 to 0.88 for daily *KT* for HelioClim-1, and are much greater than those obtained for the re-analyses both in the work of Boilley and Wald and here (Table 2). These findings of Boilley and Wald support the argument that differences in spatial support of information cannot be the only reason for the poor performances of the re-analyses.

## 5.  Conclusions

This work brings new information on the capabilities of five data sets for assessing the magnitude of the DSIS and its variability in space and time at both hourly and daily timescales in the tropical Atlantic Ocean for a more informed usage of these data in ocean sciences. Five buoys within the PIRATA network are offering enough data of high quality to perform an assessment of the two meteorological analyses MERRA-2 and ERA5 and the three satellite-derived data sets HC3v5, SARAH-2 and CRS.

It was found that the re-analyses MERRA-2 and ERA5 often report cloud-free conditions while the actual conditions are cloudy, yielding an overestimation of the DSIS in such cases. They also report actual cloud-free conditions as cloudy, yielding an underestimation. These alternating underestimations and overestimations compensate each other with a small bias as a result masking some deficiencies in properly modelling cloud properties. These conclusions are similar to those already reported regarding meteorological re-analyses as a whole (Wild, 2008). The estimates from MERRA-2 or ERA5 poorly correlate with the clearness indices at buoys: the correlation coefficient ranges between 0.48 and 0.54 for MERRA-2, and between 0.56 and 0.63 for ERA5. Hence, a large part of the variability in the optical state of the atmosphere is not captured by the MERRA-2 or ERA5 re-analyses. It is recommended not to use them in studies of the variability in time of the surface irradiance in the tropical Atlantic Ocean when it is necessary to reproduce actual measurements.

The selected moorings experience a large amount of cloud-free conditions. In these conditions, ERA5 tends to underestimate the irradiance. The bias for the two re-analyses depend on the cloud type: they exhibit more bias in irradiance in cases of medium and high level clouds than for low level clouds.

The bias varies noticeably with the calendar month, which means that MERRA-2 or ERA5 cannot be used confidently at a monthly timescale. The re-analyses exhibit small biases when compared to PIRATA measurements over one or more years though the study was limited to 4 years for 2 buoys. It can be speculated that one may use them to follow changes in yearly values of irradiance at one location.

Another striking feature is the variability of the bias and other performance indicators for both MERRA-2 and ERA5 within this ocean area which is fairly homogeneous for both the irradiance and clearness index. Accordingly, an additional recommendation on re-analyses may be not to use them to study the spatial field of irradiance at whatever timescale: the performances strongly vary from one mooring to another, especially for MERRA-2, which means that the field of surface irradiance is spatially distorted, even on a yearly timescale.

The present results bring new evidence on the qualities and limitations of MERRA-2 and ERA5 which have been little studied for the irradiance at surface. These re-analyses may be used in studies of the tropical Atlantic Ocean with proper understanding of the limitations and uncertainties. Zhao et al. (2013) proposed an empirical relationship for correcting the bias observed between MERRA estimates and measurements of monthly averages of irradiance performed at several sites in North America taking into account the dependency between the bias and $KT$ and surface elevation. The bias and the root mean square error were reduced but at the expenses of an increase in standard deviation of errors. Jones et al. (2017) have tested several methods using HC3v5 for decreasing the bias of the ERA-Interim daily estimates of $E$. They found that when compared to measurements of daily irradiance performed at 55 terrestrial stations in Europe, the bias was reduced at 10 stations and similar for the others and that the other indicators (standard deviation of errors, root mean square error, correlation coefficient, median of errors, etc.) were unchanged. Though the works were performed for MERRA or ERA-Interim, it is speculated that similar conclusions would be reached when applied to MERRA-2 or ERA5, given the similarities between these re-analyses.

The three satellite-derived data sets exhibit similar performances between them and have better performance indicators than the two re-analyses, except for the bias. The correlation coefficients are greater than 0.95 for irradiance and 0.80 for clearness index in most cases and at both hourly and daily timescales. Each data set reproduces well the dynamics of the irradiance at both timescales though amplitudes of variation in time may be hindered by the large standard deviation of the errors which amounts to approximately 80 W m⁻² for hourly DSIS and 20 W m⁻² for daily DSIS. The same conclusion applies to the clearness index.

The three satellite-derived data sets exhibit overestimation, with the lowest biases reached by CRS and ranging between 11 and 37 W m⁻² depending on the mooring. The bias for SARAH-2 shows a tendency to decrease as the mean clearness index increases. The biases for HelioClim-3v5 and CRS are almost the same at the five moorings and the irradiance pattern may not be noticeably distorted by artefacts induced by spatial changes in bias. This suggests that HC3v5 and CRS may be used confidently when the study of the irradiance field and of its spatial features is at stake.

It can be concluded that the three satellite-derived data sets are appropriate to study the dynamics of the downward solar irradiance at the surface of the tropical Atlantic Ocean and that their performances are fairly similar. Assuming that pyranometer measurements of the PIRATA buoys achieve the "moderate quality" defined by WMO (2008, rev. 2012), one may ask if the estimates from the satellite-derived data sets are compliant with "moderate quality", assuming that the bias may be removed. The relative uncertainty is defined as the 95 % probability (P95) and should not exceed 20 % to meet the "moderate quality" (WMO, 2008, rev. 2012). The total uncertainty takes into account the uncertainty of PIRATA and the uncertainty of the estimates. It can be expressed in a first approximation as the quadratic sum of both uncertainties. As a consequence, the total relative uncertainty should not exceed 28 % (P95), or 14 % (P66) if the estimates were of "moderate" quality. The standard deviations (P66) for each data set reported in Table 3 are below 14 %. It can be concluded that under this approximation, the three satellite-derived data sets can be considered of moderate quality if bias can be removed.

One may note several similarities in performances between HC3v5 and SARAH-2. It is speculated that this is partly due to the fact that they exploit the same method, Heliosat-2, though the implementation differs.

Other data sets are available that cover the tropical Atlantic Ocean and may be assessed against the PIRATA measurements to gain knowledge on their limitations and confidence in their use. Examples are the satellite-derived OSI-SAF (www.osi-saf.org) or the Japanese 55-year re-analysis (JRA 55, Kang et al., 2015; Kobayashi et al., 2015).

The findings reported here are consistent with the very few works already published for terrestrial stations. This demonstrates a posteriori that the PIRATA measurements may be used for the validation of models and data sets. However, some uncertainties remain. Potential biases in the PIRATA time series are an issue and difficult to estimate. It complicates validation of the satellite-derived data sets. While the different levels and variability of surface irradiance at buoys might impact the quality of the satellite-based data sets, a reduced data quality of the buoy data (despite the quality control applied) might also have an impact on the presented evaluation. Studies like these when multiple data records are considered can help to identify problems in surface reference measurements (Urraca et al., 2017). The PIRATA network is a unique and valuable means to study and monitor the surface irradiance in the tropical Atlantic Ocean and deserves support for operations to further enrich the data records.

## 6. Data availability

PIRATA measurements performed every 2 min were downloaded from the web site (www.pmel.noaa.gov/tao/drupal/disdel/) of the National Oceanic and Atmospheric Administration (NOAA) of the U.S.A. The authors acknowledge the help of the GTMBA Project Office of NOAA/PMEL in getting the data and the PIRATA team for servicing the network and freely providing the data.

Time-series of HelioClim-3v5 data were downloaded from the SoDa Service web site (www.soda-pro.com) managed by the company Transvalor. Data are available to anyone for free for years 2004-2006 as a GEOSS Data-CORE (GEOSS Data Collection of Open Resources for Everyone) and for a charge for the most recent years with the amount depending on requests and requester. The time-series used in this article are available for free in CSV format by request to Mireille Lefèvre.

Time-series of SARAH-2 data were extracted from the gridded data sets available at https://doi.org/10.5676/EUM_SAF_CM/SARAH/V002.

Time-series of CAMS Radiation Service data were downloaded from the SoDa Service web site (www.soda-pro.com).

Time-series of cloud classification were downloaded from the SoDa Service web site (www.soda-pro.com).

MERRA-2 times-series were extracted from the gridded data sets available at https://goldsmr4.gesdisc.eosdis.nasa.gov/data/MERRA2/.

ERA5 times-series were extracted from the gridded data sets available at http://apps.ecmwf.int/data-catalogues/era5/?class=ea&stream=enda&expver=1.

## 7. Author contribution

All authors contributed equally to this work.

## 8. Competing interests

The authors declare no competing interests.

## 9. Disclaimer

N/A

## 10. Acknowledgements

The research leading to these results has partly received funding from the Copernicus Atmosphere Monitoring Service, a program being operated by the European Centre for Medium-Range Weather Forecasts (ECMWF) on behalf of the European Union. The authors thank the French company Transvalor for taking care of the SoDa Service for the common good, thus providing an efficient access to the HelioClim databases. The authors thank especially Gregory Foltz for his helpful advice on the PIRATA measurements, the two anonymous reviewers and Yehia Eissa for their kind and useful comments which greatly help in improving the clarity of this article.

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

# Appendix

# (0°N, 0°E)

### 11.1. Satellite-derived data sets

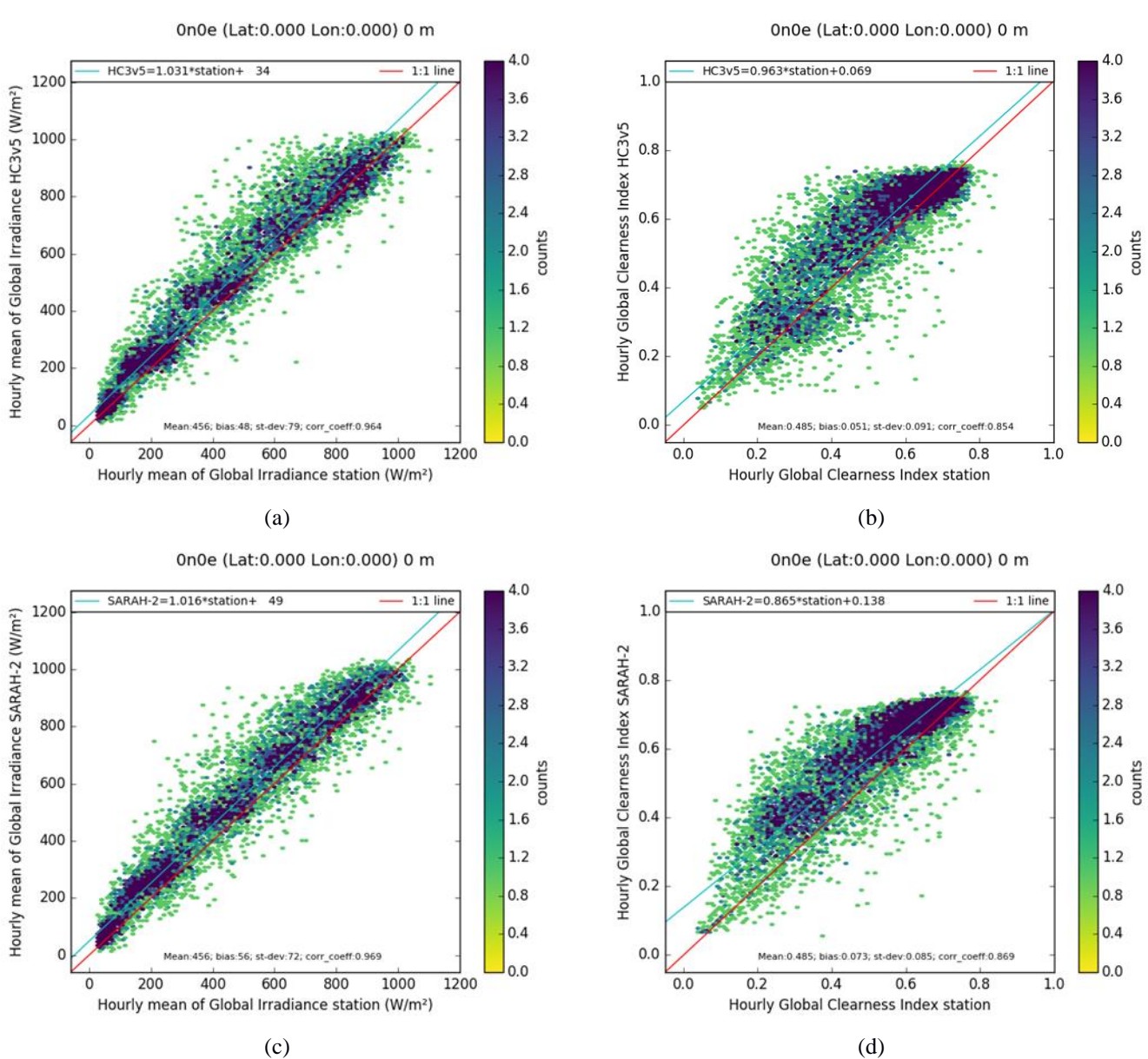

(a)                  (b)

(c)                  (d)

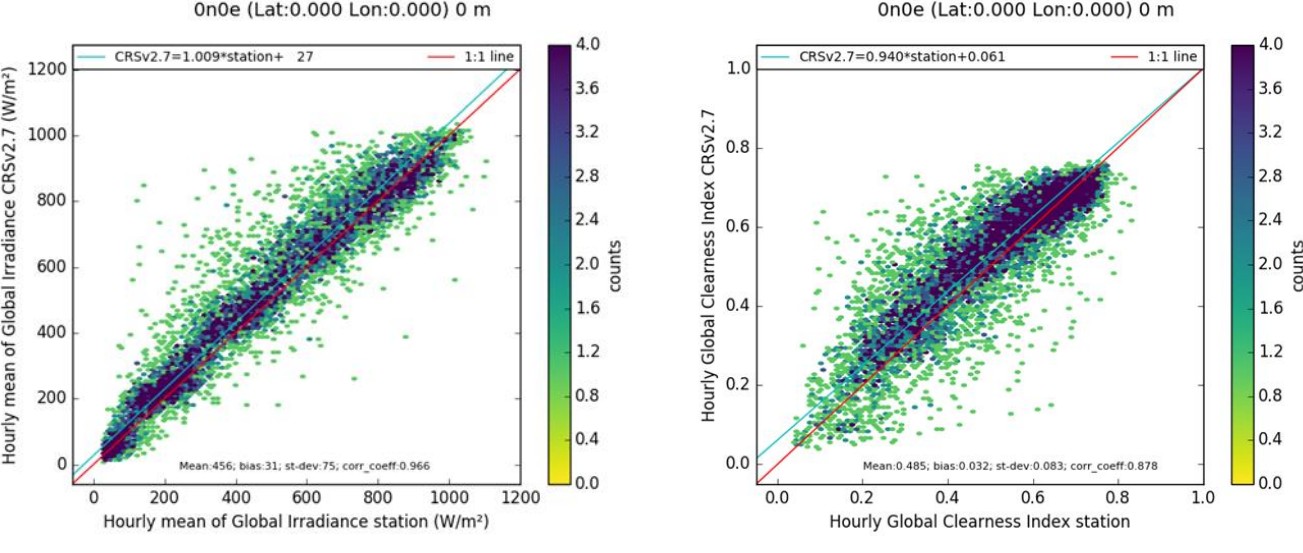

(e)                                                              (f)

**2D histogram of PIRATA measurements (horizontal axis) and data sets (vertical axis) at (0°N, 0°E) for *E* and *KT*. HC3v5: (a) and (b); SARAH-2: (c) and (d); CRS: (e) and (f). Ideally, the dots should lie along the red line (1:1 line). The blue line is the affine function fitted over the points and should ideally overlay the red line.**

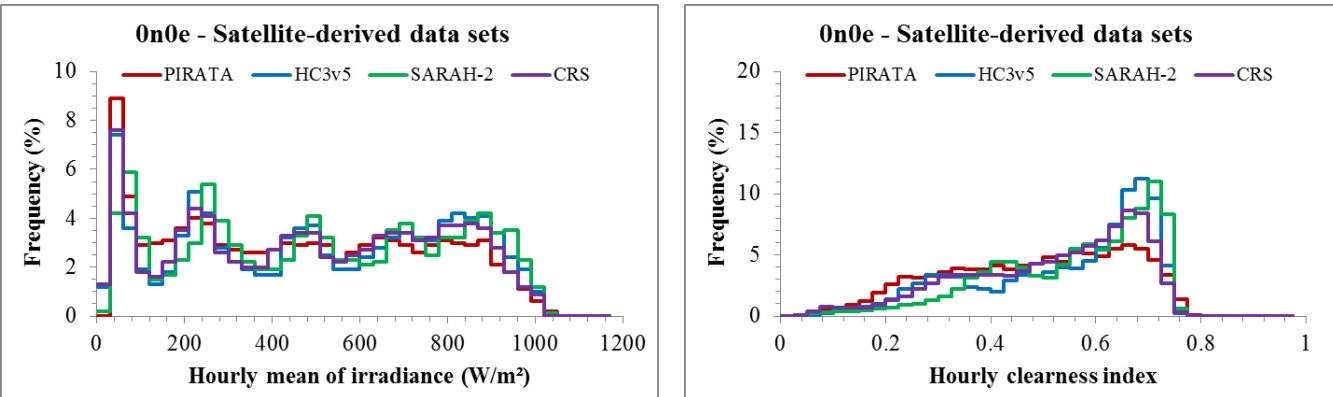

5  **Frequency distributions of PIRATA measurements (red) and data sets (HC3v5: blue, SARAH-2: green, CRS: purple) at (0°N, 0°E) for *E* (left) and *KT* (right). If the coloured line is above, respectively below, the red one for a given sub-range of values, it means that the data set produces these values too frequently, respectively too rarely with respect to the PIRATA measurements.**

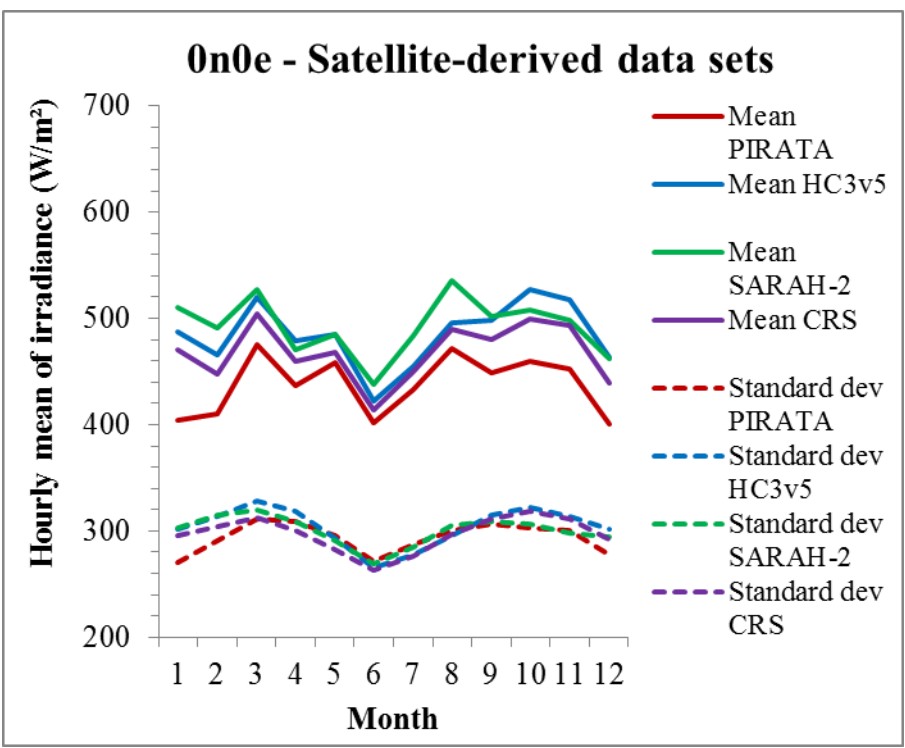

**Monthly means (line) and standard deviations (dotted line) of hourly DSIS, in W m⁻², from PIRATA measurements (red) and data sets (HC3v5: blue, SARAH-2: green, CRS: purple) at (0°N, 0°E). A difference between red line (measurements) and coloured line (data set) for a given month denotes a systematic error for this month: underestimation if the coloured line is below the red line, overestimation otherwise. For a given month, a coloured dotted line above the red one means that the data set produces too much variability for this month; in the opposite case, the data set does not contain enough variability.**

## 11.2.    Re-analysis data sets

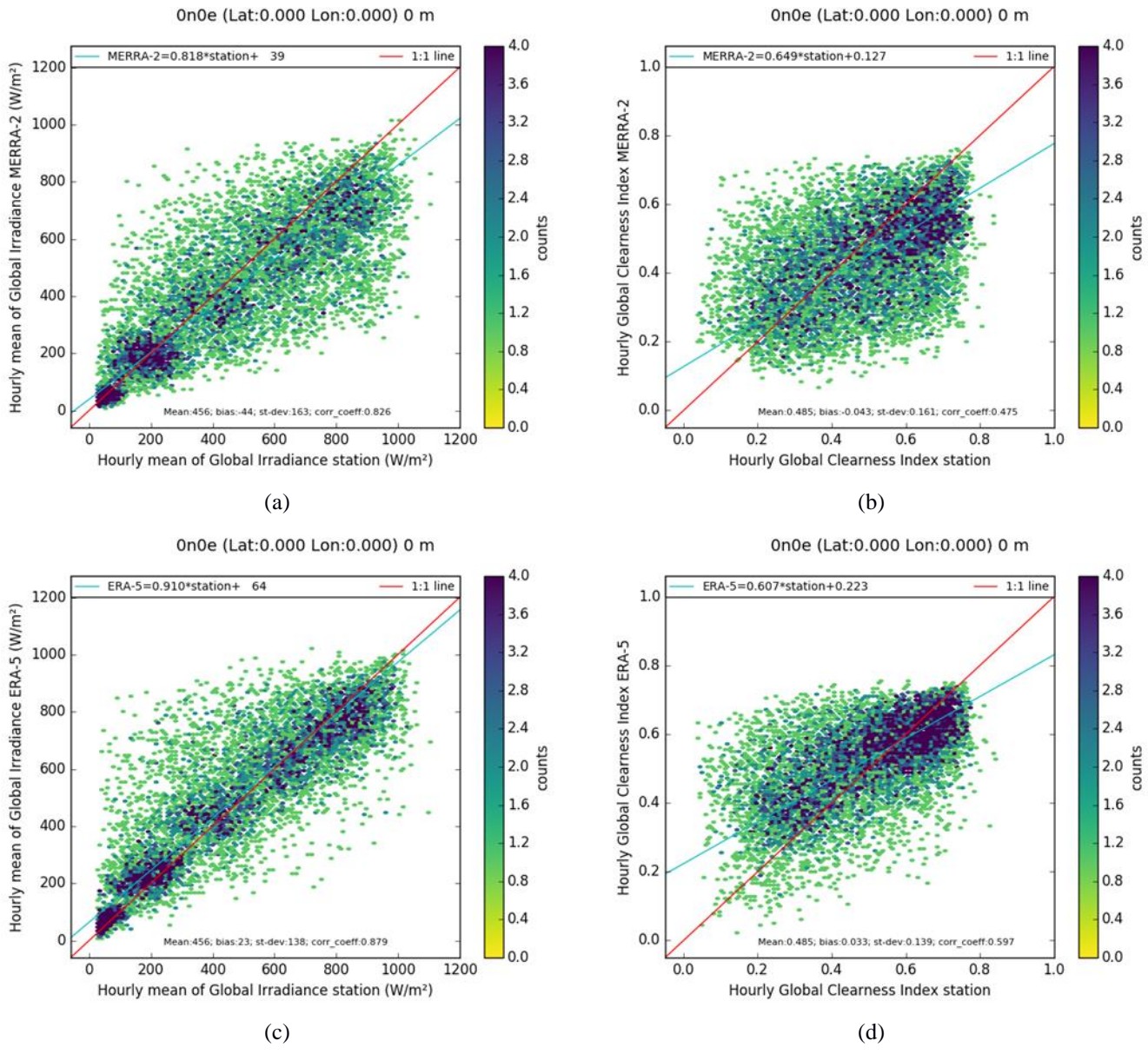

(a)

(b)

(c)

(d)

**2D histogram of PIRATA measurements (horizontal axis) and data sets (vertical axis) at (0°N, 0°E) for *E* and *KT*. MERRA-2: (a) and (b); ERA5: (c) and (d). Ideally, the dots should lie along the red line (1:1 line). The blue line is the affine function fitted over the points and should ideally overlay the red line.**

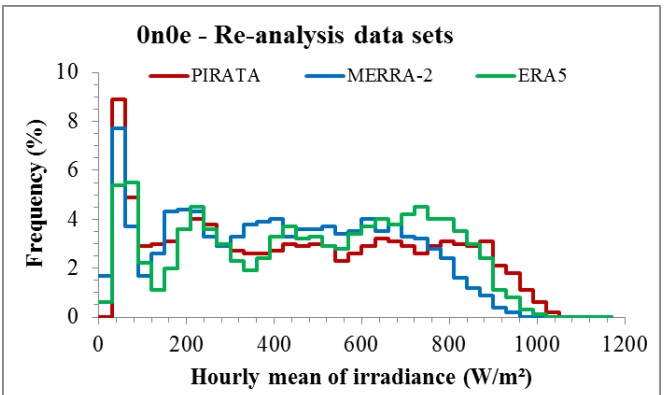
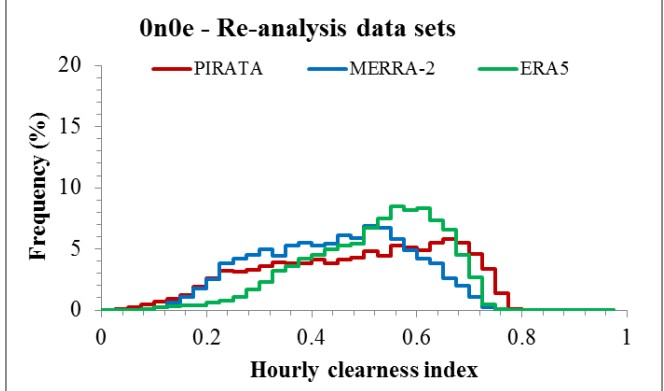

Frequency distributions of PIRATA measurements (red) and data sets (MERRA-2: blue, ERA5: green) at (0°N, 0°E) for *E* (left) and *KT* (right). If the coloured line is above, respectively below, the red one for a given sub-range of values, it means that the data set produces these values too frequently, respectively too rarely with respect to the PIRATA measurements.

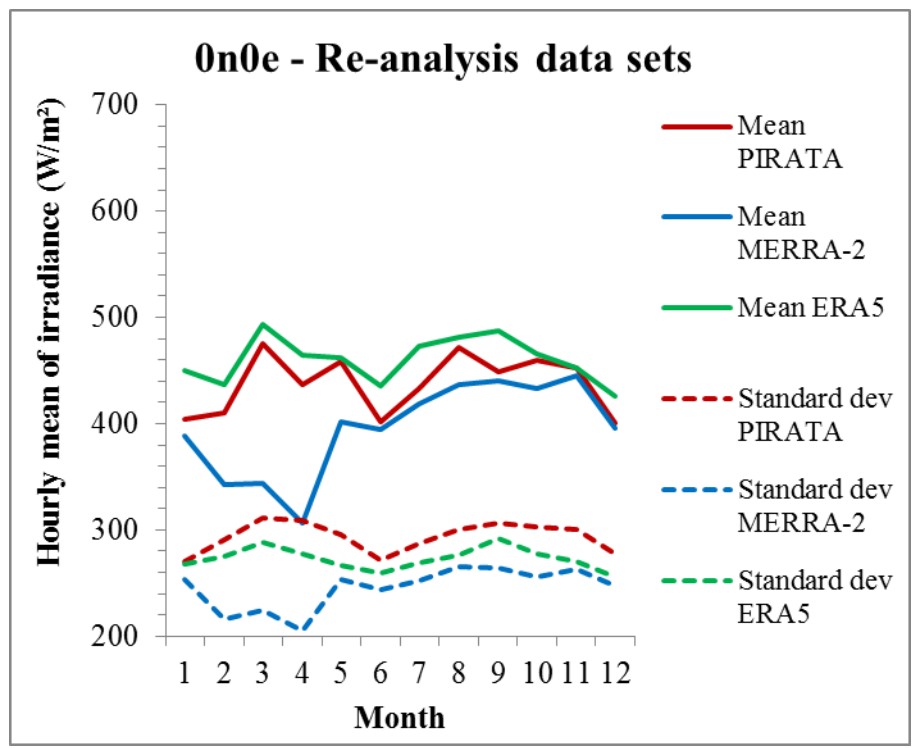

Monthly means (line) and standard deviations (dotted line) of hourly DSIS, in W m$^{-2}$, from PIRATA measurements (red) and data sets (MERRA-2: blue, ERA5: green) at (0°N, 0°E). A difference between red line (measurements) and coloured line (data set) for a given month denotes a systematic error for this month: underestimation if the coloured line is below the red line, overestimation otherwise. For a given month, a coloured dotted line above the red one means that the data set produces too much variability for

10 this month; in the opposite case, the data set does not contain enough variability.

## 11.3. Satellite-derived data sets

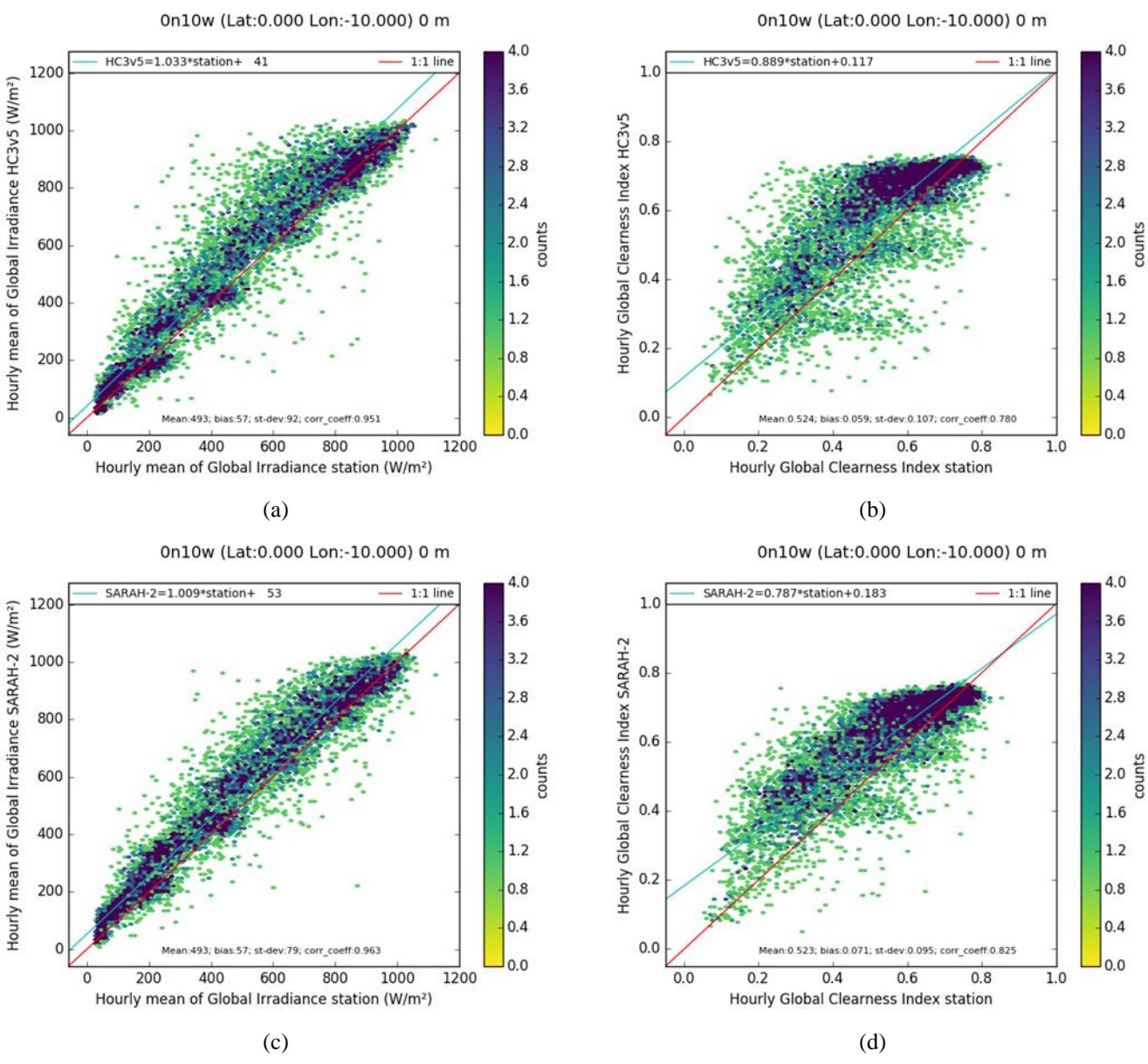

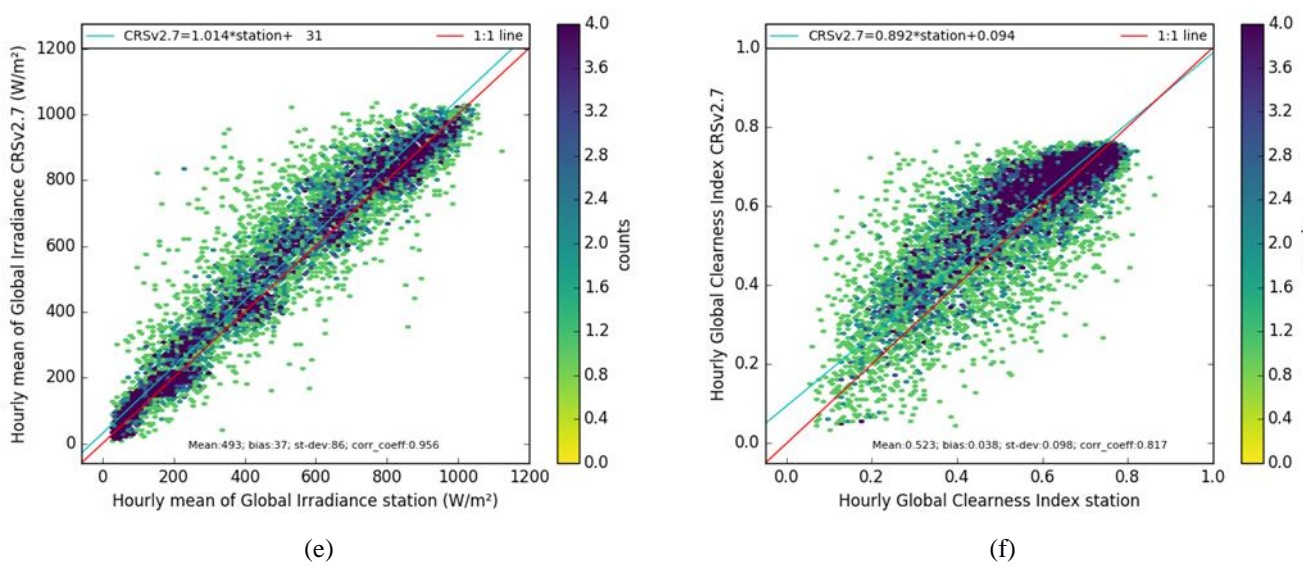

(e)  (f)

**2D histogram of PIRATA measurements (horizontal axis) and data sets (vertical axis) at (0°N, 10°W) for *E* and *KT*. HC3v5: (a) and (b); SARAH-2: (c) and (d); CRS: (e) and (f). Ideally, the dots should lie along the red line (1:1 line). The blue line is the affine function fitted over the points and should ideally overlay the red line.**

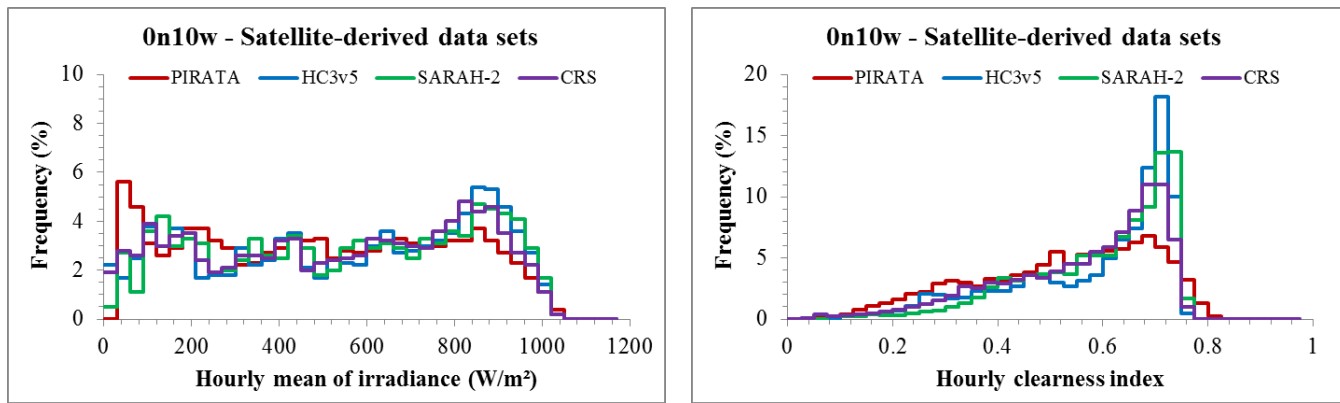

5  **Frequency distributions of PIRATA measurements (red) and data sets (HC3v5: blue, SARAH-2: green, CRS: purple) at (0°N, 10°W) for *E* (left) and *KT* (right). If the coloured line is above, respectively below, the red one for a given sub-range of values, it means that the data set produces these values too frequently, respectively too rarely with respect to the PIRATA measurements.**

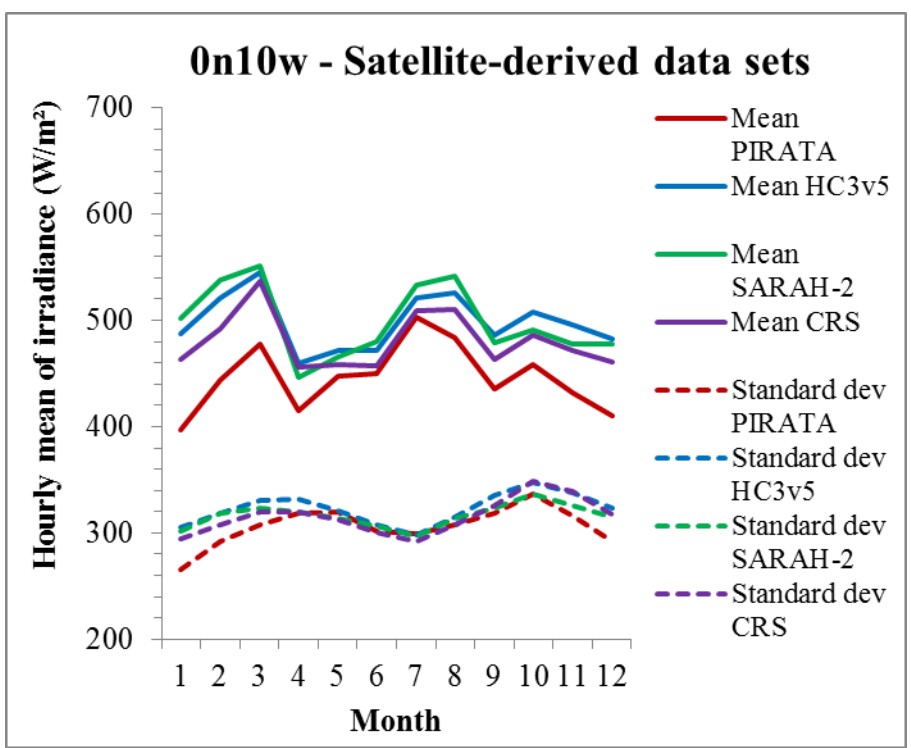

Monthly means (line) and standard deviations (dotted line) of hourly DSIS, in W m$^{-2}$, from PIRATA measurements (red) and data sets (HC3v5: blue, SARAH-2: green, CRS: purple) at (0°N, 10°W). A difference between red line (measurements) and coloured line (data set) for a given month denotes a systematic error for this month: underestimation if the coloured line is below the red line, overestimation otherwise. For a given month, a coloured dotted line above the red one means that the data set produces too much variability for this month; in the opposite case, the data set does not contain enough variability.

## 11.4. Re-analysis data sets

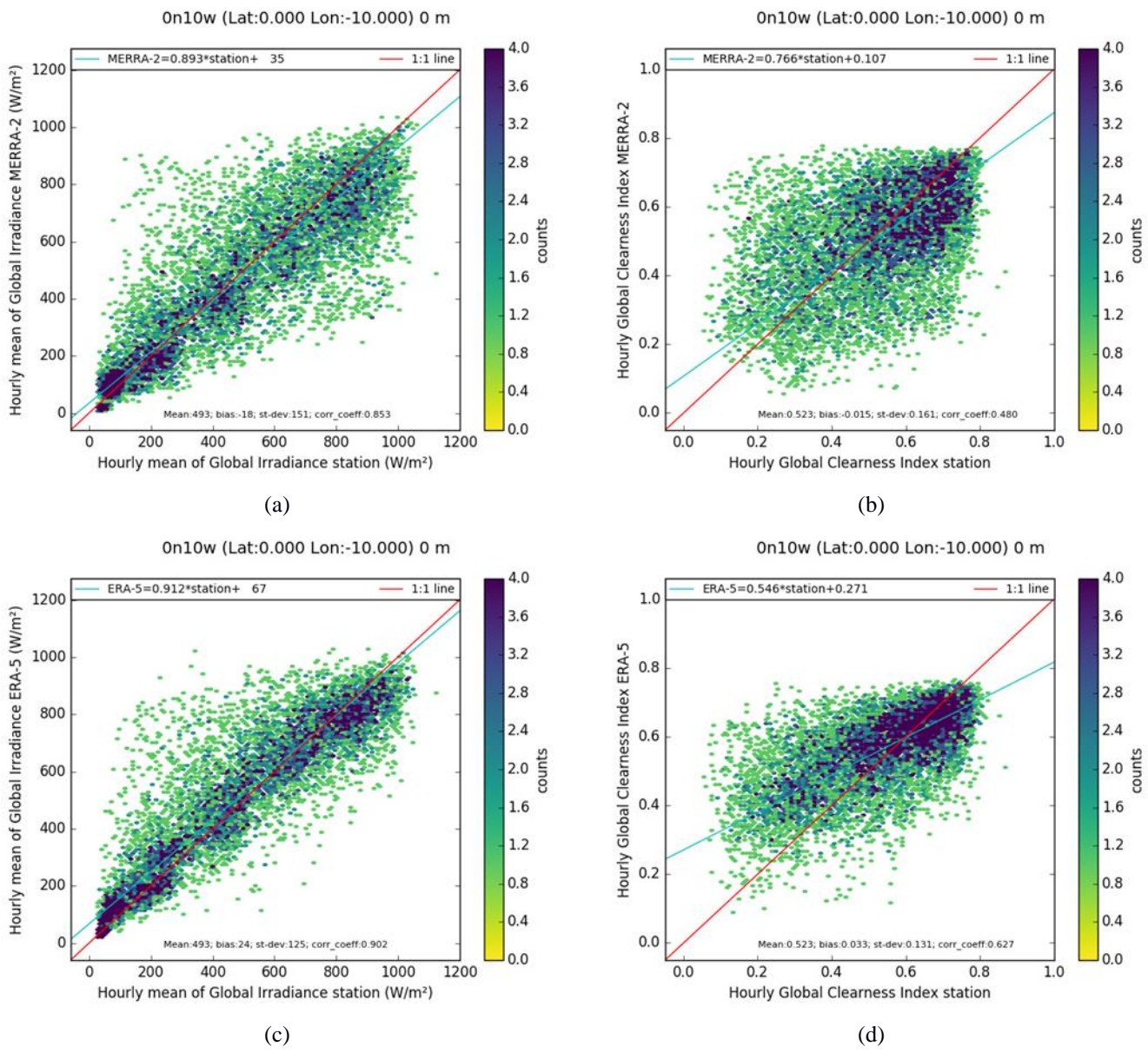

**2D histogram of PIRATA measurements (horizontal axis) and data sets (vertical axis) at (0°N, 10°W) for *E* and *KT*. MERRA-2: (a) and (b); ERA5: (c) and (d). Ideally, the dots should lie along the red line (1:1 line). The blue line is the affine function fitted over the points and should ideally overlay the red line.**

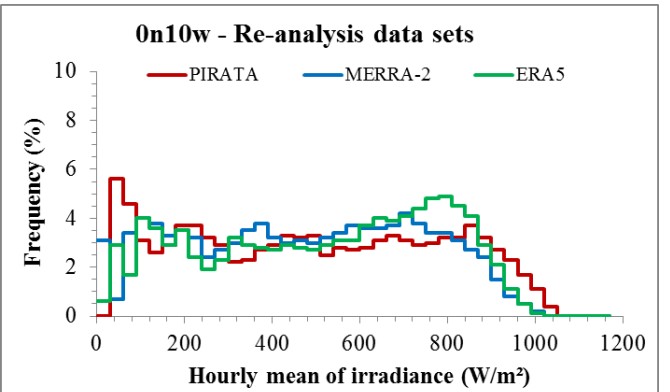
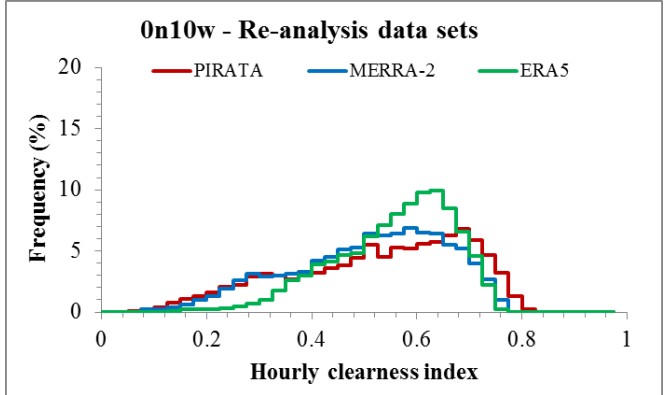

**Frequency distributions of PIRATA measurements (red) and data sets (MERRA-2: blue, ERA5: green) at (0°N, 10°W) for *E* (left) and *KT* (right). If the coloured line is above, respectively below, the red one for a given sub-range of values, it means that the data set produces these values too frequently, respectively too rarely with respect to the PIRATA measurements.**

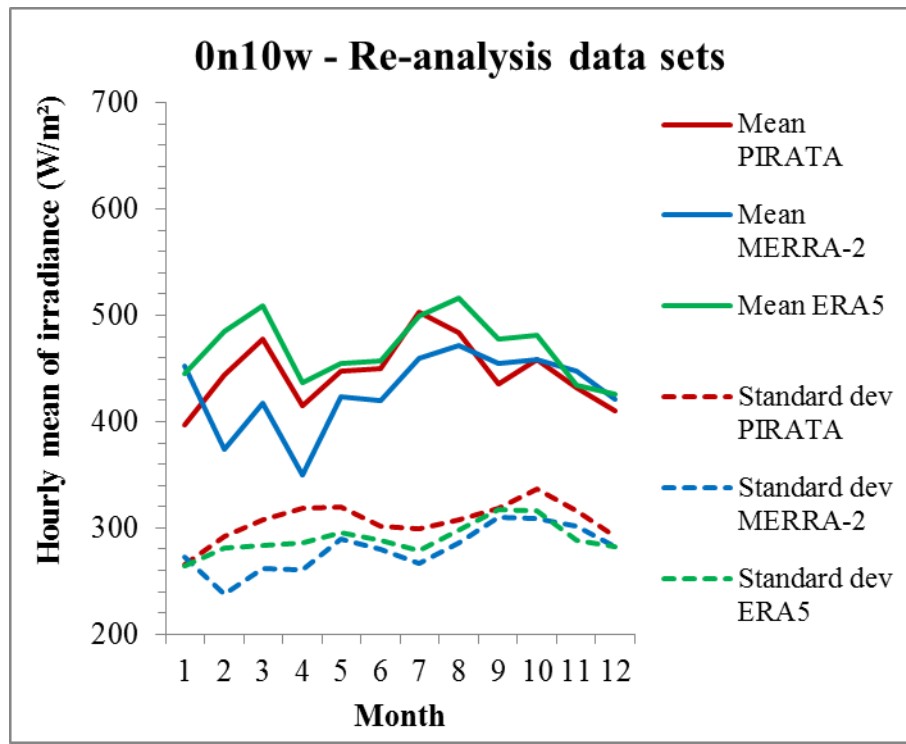

**Monthly means (line) and standard deviations (dotted line) of hourly DSIS, in W m⁻², from PIRATA measurements (red) and data sets (MERRA-2: blue, ERA5: green) at (0°N, 10°W). A difference between red line (measurements) and coloured line (data set) for a given month denotes a systematic error for this month: underestimation if the coloured line is below the red line, overestimation otherwise. For a given month, a coloured dotted line above the red one means that the data set produces too much variability for this month; in the opposite case, the data set does not contain enough variability.**

# (0°N, 23°W)

## 11.5. Satellite-derived data sets

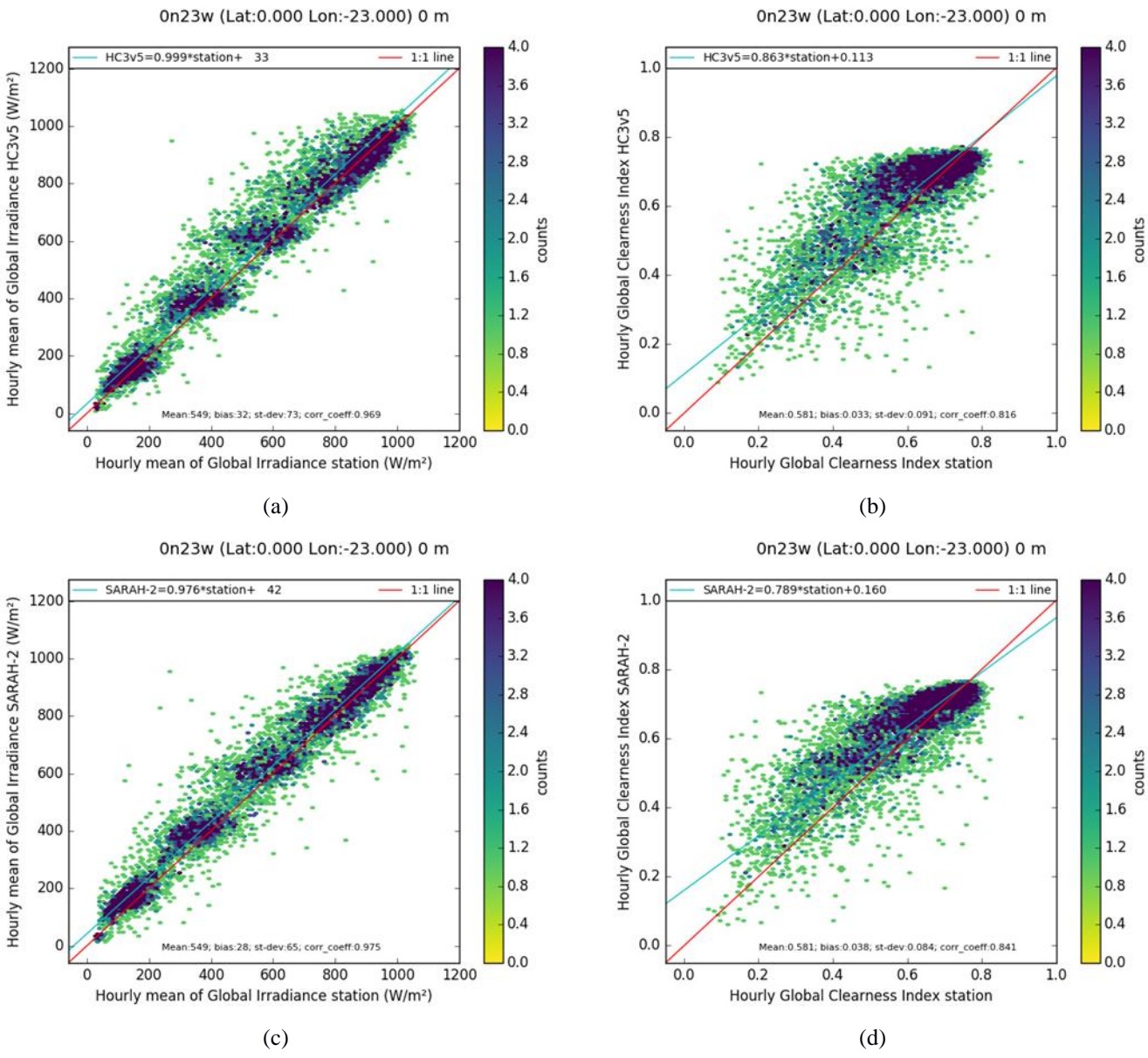

(a)

(b)

(c)

(d)

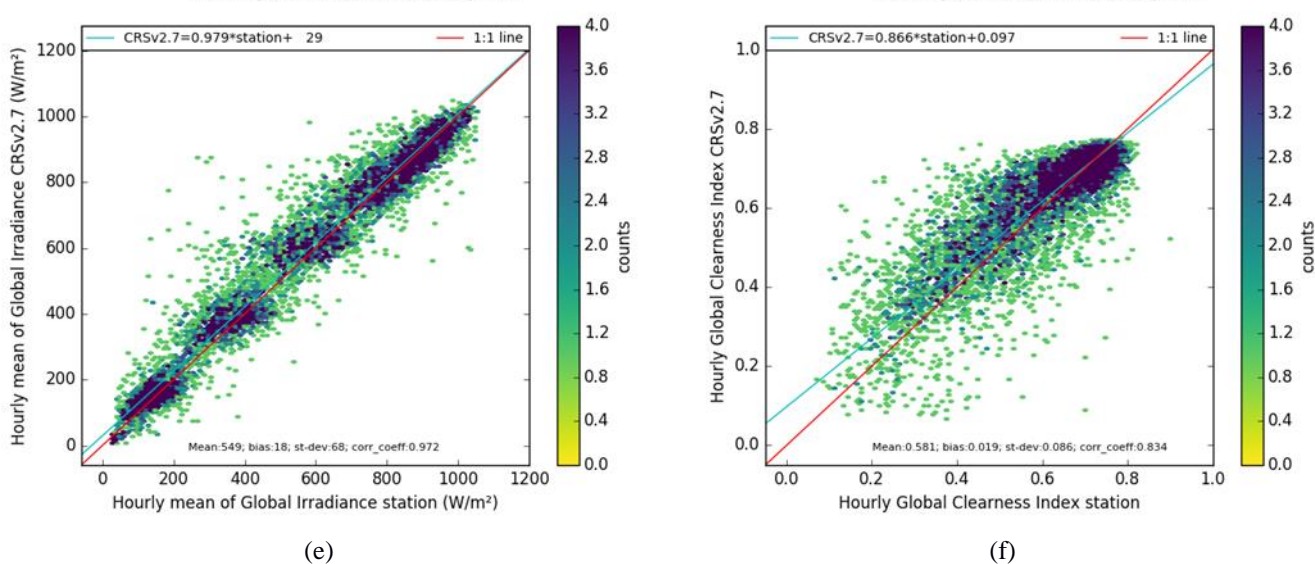

(e) (f)

**2D histogram of PIRATA measurements (horizontal axis) and data sets (vertical axis) at (0°N, 23°W) for *E* and *KT*. HC3v5: (a) and (b); SARAH-2: (c) and (d); CRS: (e) and (f). Ideally, the dots should lie along the red line (1:1 line). The blue line is the affine function fitted over the points and should ideally overlay the red line.**

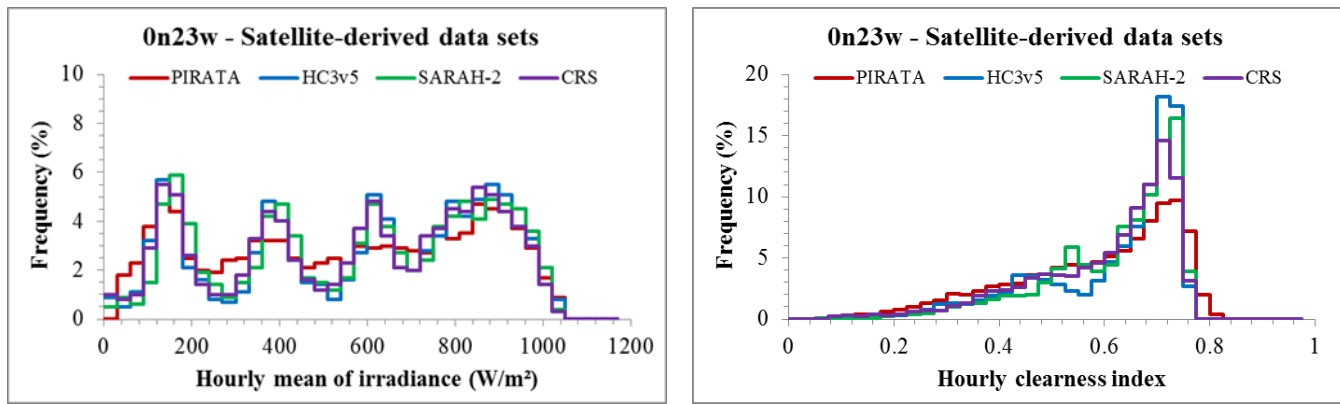

5     **Frequency distributions of PIRATA measurements (red) and data sets (HC3v5: blue, SARAH-2: green, CRS: purple) at (0°N, 23°W) for *E* (left) and *KT* (right). If the coloured line is above, respectively below, the red one for a given sub-range of values, it means that the data set produces these values too frequently, respectively too rarely with respect to the PIRATA measurements.**

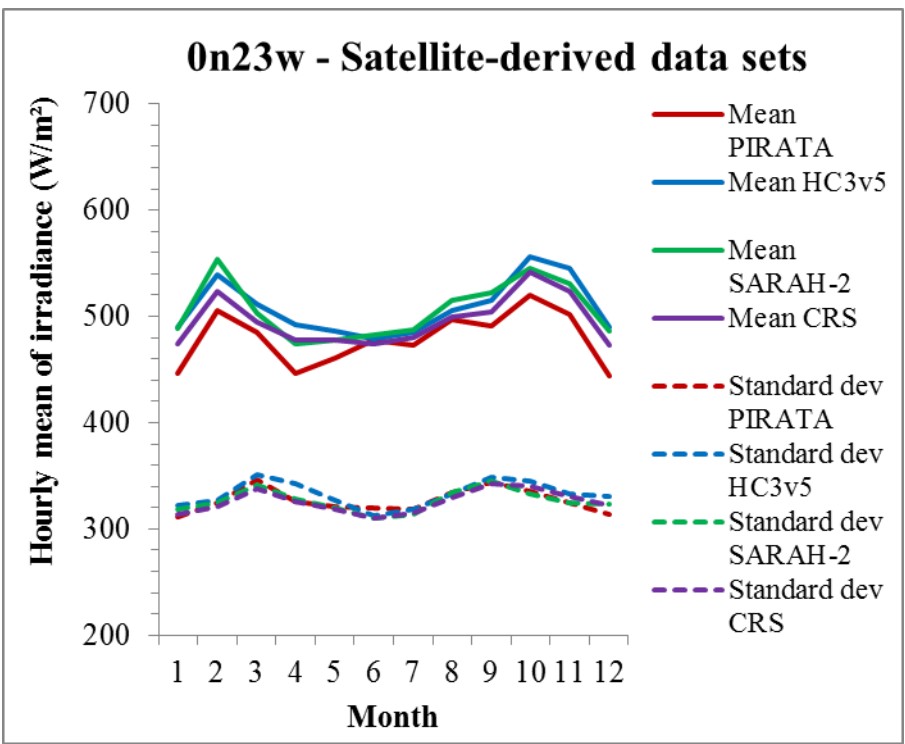

Monthly means (line) and standard deviations (dotted line) of hourly DSIS, in W m$^{-2}$, from PIRATA measurements (red) and data sets (HC3v5: blue, SARAH-2: green, CRS: purple) at (0°N, 23°W). A difference between red line (measurements) and coloured line (data set) for a given month denotes a systematic error for this month: underestimation if the coloured line is below the red line, overestimation otherwise. For a given month, a coloured dotted line above the red one means that the data set produces too much variability for this month; in the opposite case, the data set does not contain enough variability.

## 11.6. Re-analysis data sets

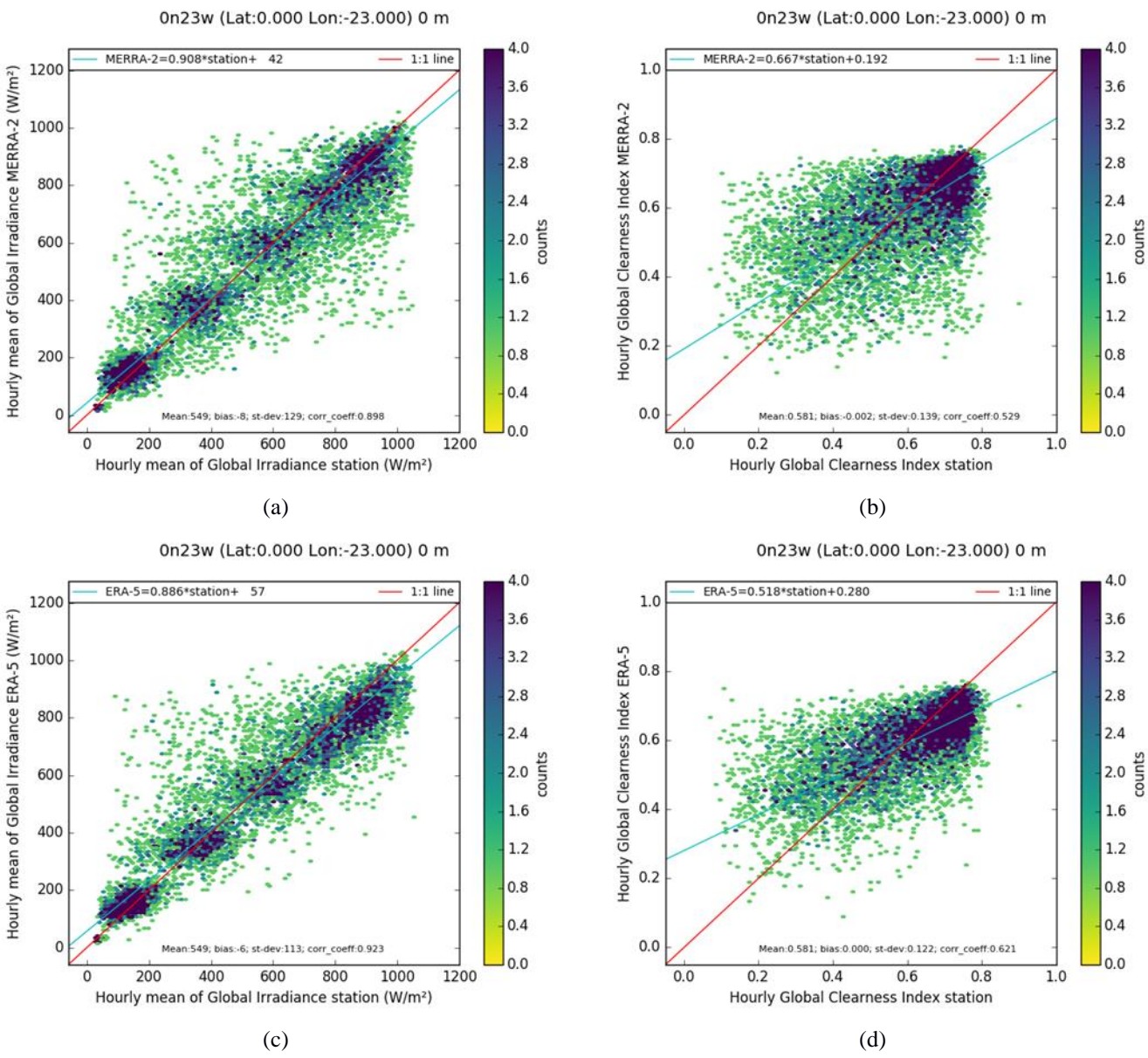

**2D histogram of PIRATA measurements (horizontal axis) and data sets (vertical axis) at (0°N, 23°W) for *E* and *KT*. MERRA-2: (a) and (b); ERA5: (c) and (d). Ideally, the dots should lie along the red line (1:1 line). The blue line is the affine function fitted over the points and should ideally overlay the red line.**

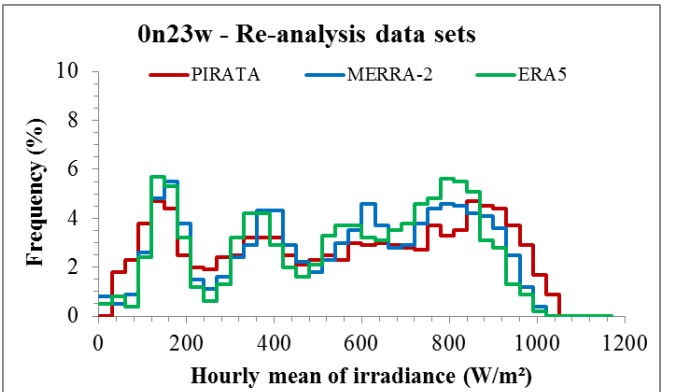 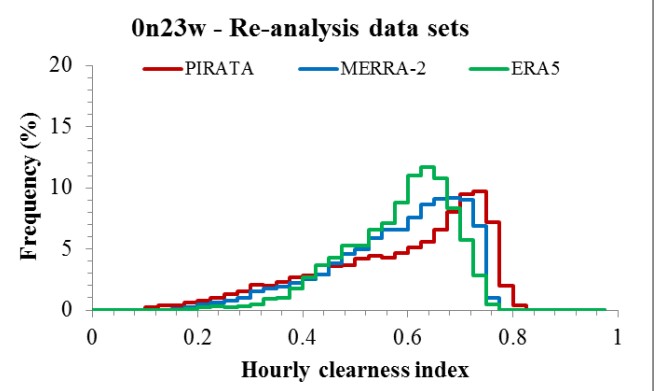

**Frequency distributions of PIRATA measurements (red) and data sets (MERRA-2: blue, ERA5: green) at (0°N, 23°W) for *E* (left) and *KT* (right). If the coloured line is above, respectively below, the red one for a given sub-range of values, it means that the data set produces these values too frequently, respectively too rarely with respect to the PIRATA measurements.**

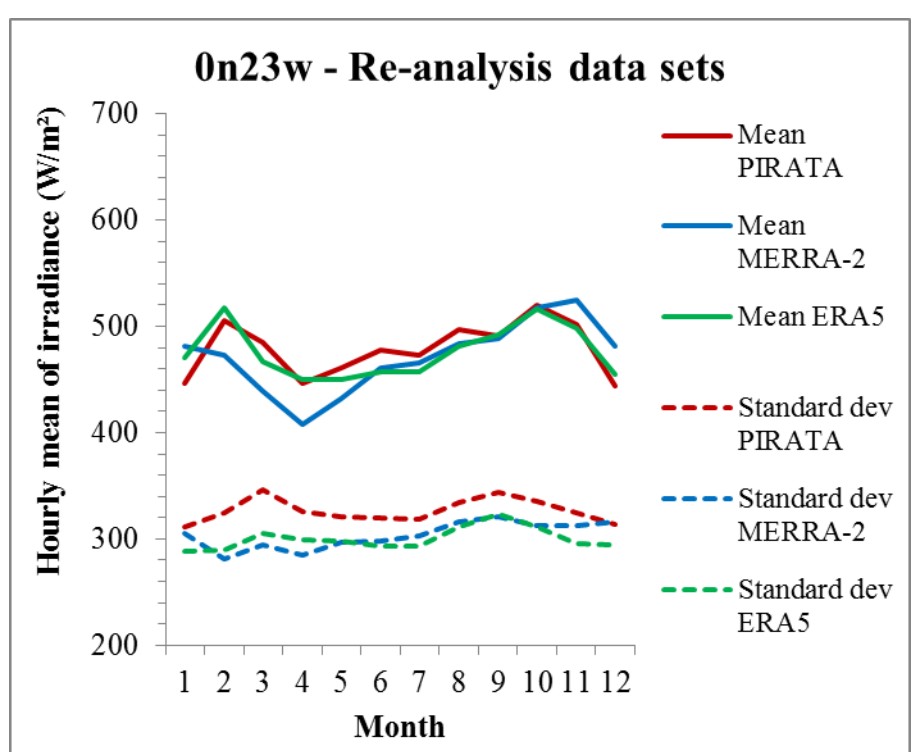

**Monthly means (line) and standard deviations (dotted line) of hourly DSIS, in W m⁻², from PIRATA measurements (red) and data sets (MERRA-2: blue, ERA5: green) at (0°N, 23°W). A difference between red line (measurements) and coloured line (data set) for a given month denotes a systematic error for this month: underestimation if the coloured line is below the red line, overestimation otherwise. For a given month, a coloured dotted line above the red one means that the data set produces too much variability for**
10 **this month; in the opposite case, the data set does not contain enough variability.**

## 11.7.    Satellite-derived data sets

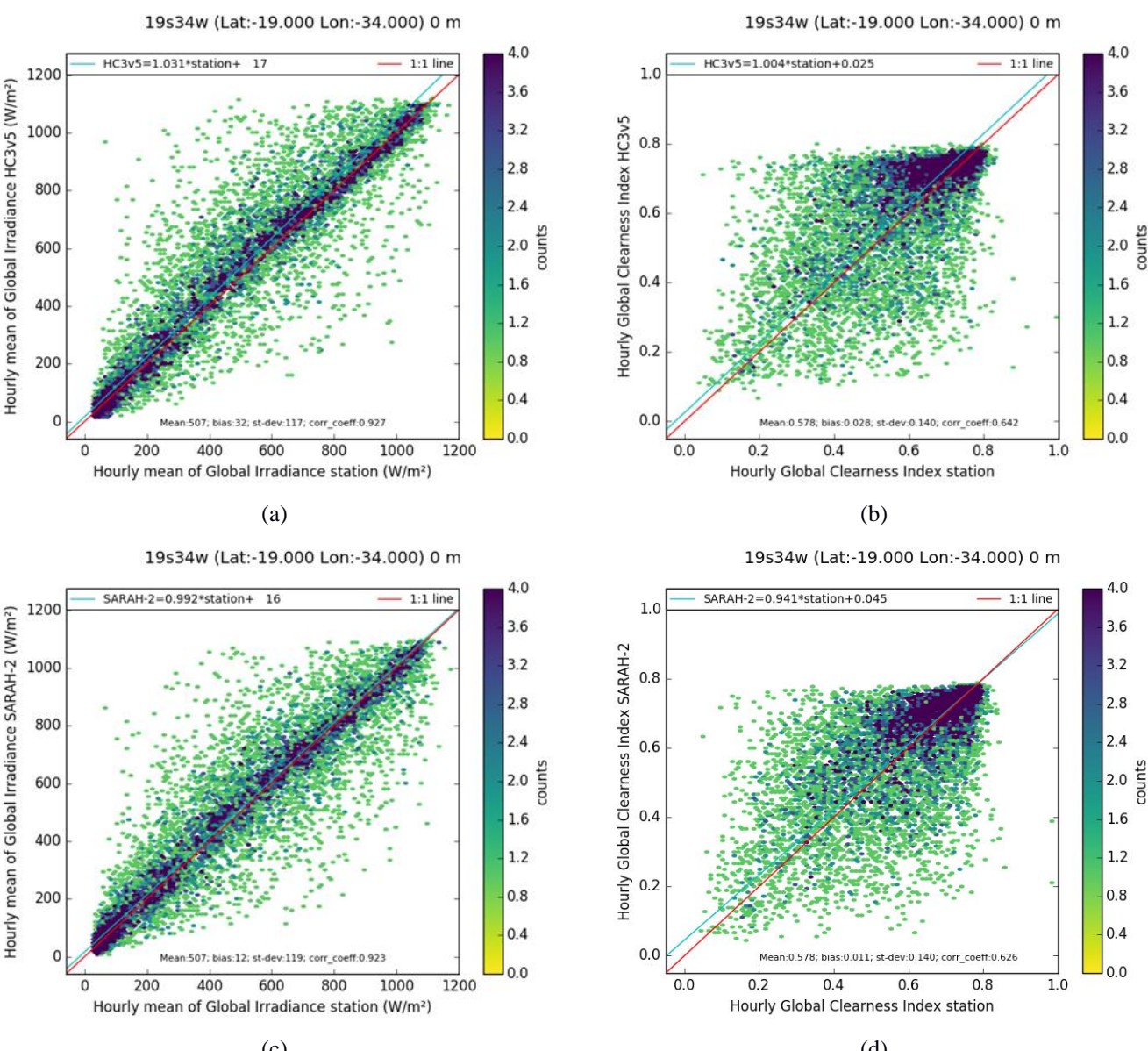

(a)                                                     (b)

(c)                                                     (d)

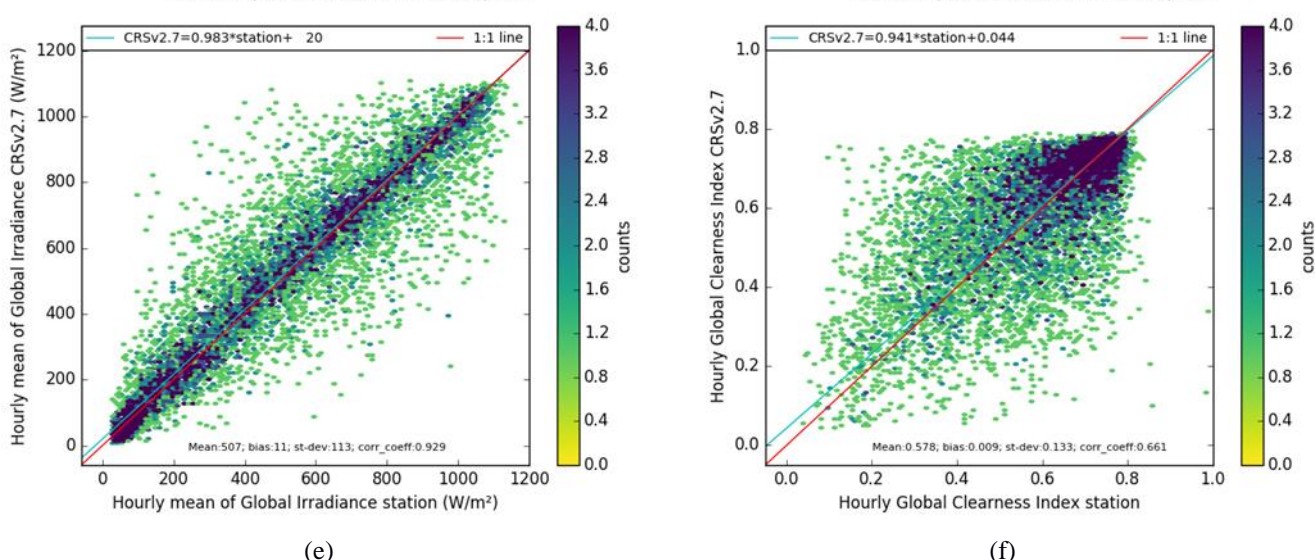

(e)                                                    (f)

**2D histogram of PIRATA measurements (horizontal axis) and data sets (vertical axis) at (19°S, 34°W) for *E* and *KT*. HC3v5: (a) and (b); SARAH-2: (c) and (d); CRS: (e) and (f). Ideally, the dots should lie along the red line (1:1 line). The blue line is the affine function fitted over the points and should ideally overlay the red line.**

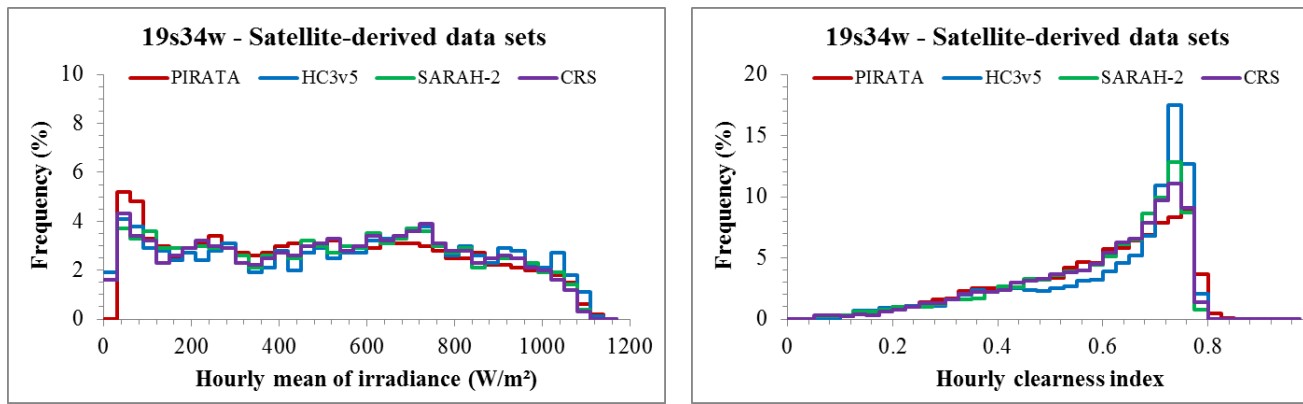

5      **Frequency distributions of PIRATA measurements (red) and data sets (HC3v5: blue, SARAH-2: green, CRS: purple) at (19°S, 34°W) for *E* (left) and *KT* (right). If the coloured line is above, respectively below, the red one for a given sub-range of values, it means that the data set produces these values too frequently, respectively too rarely with respect to the PIRATA measurements.**

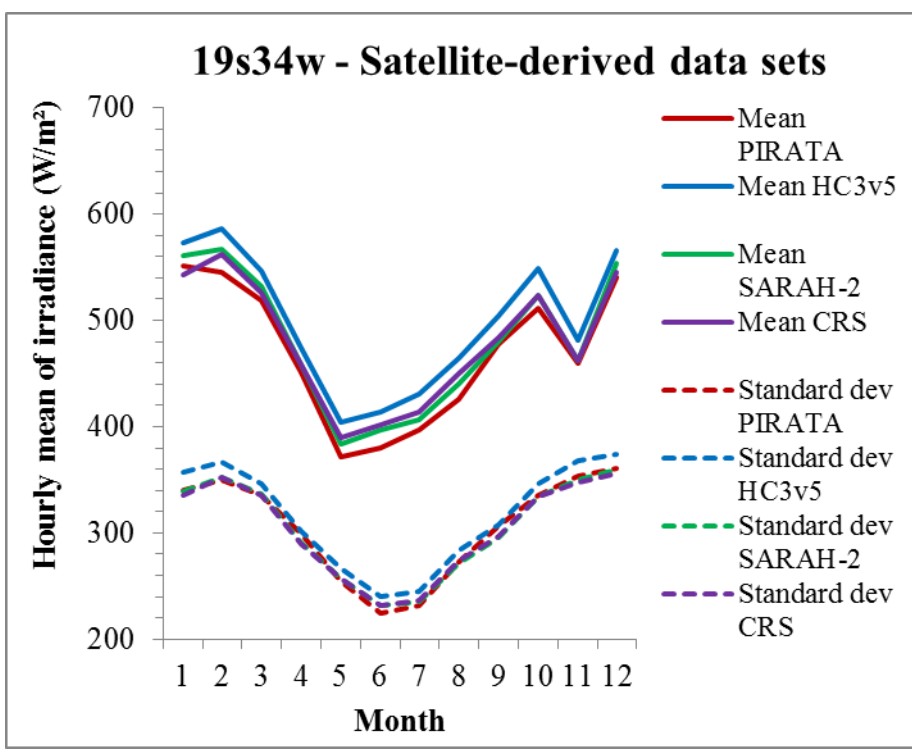

Monthly means (line) and standard deviations (dotted line) of hourly DSIS, in W m$^{-2}$, from PIRATA measurements (red) and data sets (HC3v5: blue, SARAH-2: green, CRS: purple) at (19°S, 34°W). A difference between red line (measurements) and coloured line (data set) for a given month denotes a systematic error for this month: underestimation if the coloured line is below the red line, overestimation otherwise. For a given month, a coloured dotted line above the red one means that the data set produces too much variability for this month; in the opposite case, the data set does not contain enough variability.

## 11.8.  Re-analysis data sets

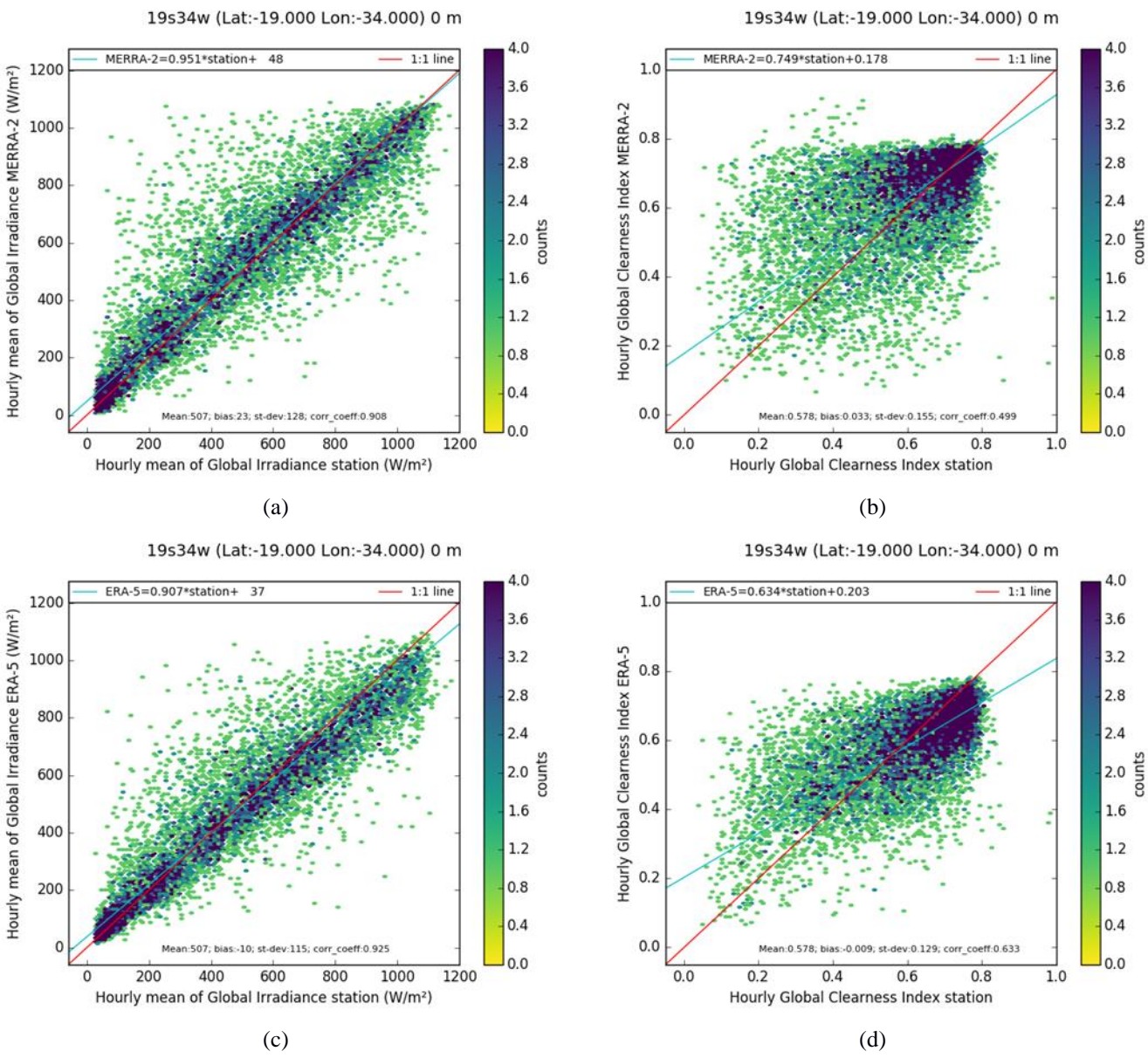

**2D histogram of PIRATA measurements (horizontal axis) and data sets (vertical axis) at (19°S, 34°W) for *E* and *KT*. MERRA-2: (a) and (b); ERA5: (c) and (d). Ideally, the dots should lie along the red line (1:1 line). The blue line is the affine function fitted over the points and should ideally overlay the red line.**

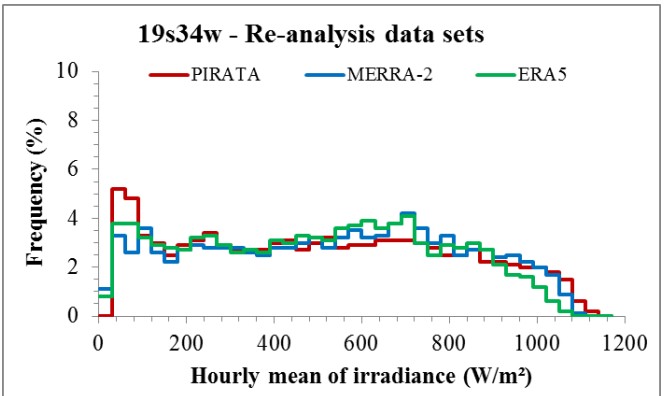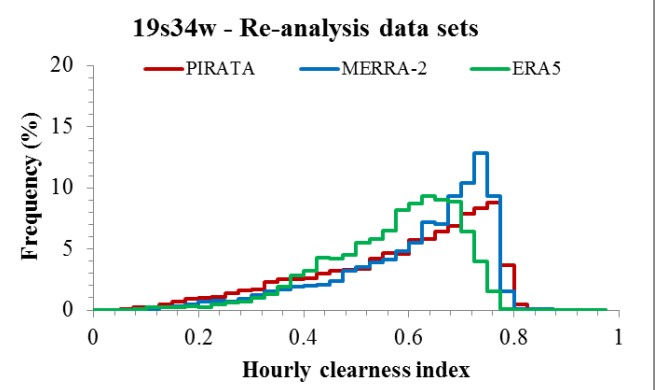

**Frequency distributions of PIRATA measurements (red) and data sets (MERRA-2: blue, ERA5: green) at (19°S, 34°W) for *E* (left) and *KT* (right). If the coloured line is above, respectively below, the red one for a given sub-range of values, it means that the data set produces these values too frequently, respectively too rarely with respect to the PIRATA measurements.**

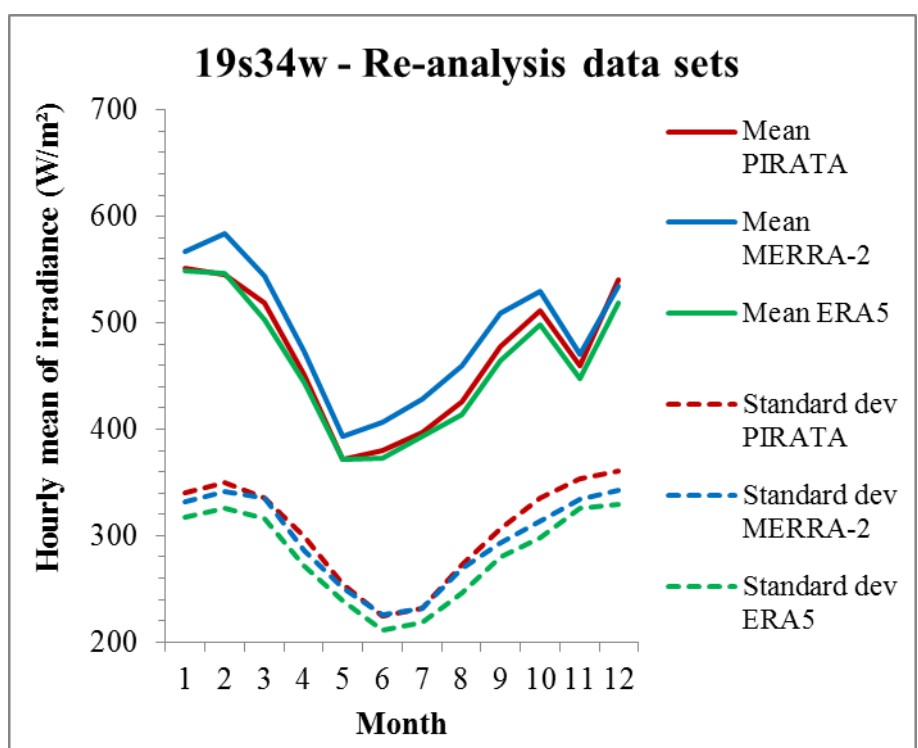

**Monthly means (line) and standard deviations (dotted line) of hourly DSIS, in W m⁻², from PIRATA measurements (red) and data sets (MERRA-2: blue, ERA5: green) at (19°S, 34°W). A difference between red line (measurements) and coloured line (data set) for a given month denotes a systematic error for this month: underestimation if the coloured line is below the red line, overestimation otherwise. For a given month, a coloured dotted line above the red one means that the data set produces too much**

10  **variability for this month; in the opposite case, the data set does not contain enough variability.**