# Peer review of "Estimating downwelling solar irradiance at the surface of the tropical Atlantic Ocean: A comparison of PIRATA measurements against several re-analyses and satellite-derived data sets"

_Ocean Science, 2017_

## Referee Comment (RC1) · Anonymous Referee #1 · 17 Jan 2018

Review of "Estimating downwelling solar irradiance at the surface of the tropical Atlantic Ocean: A comparison of PIRATA measurements against several re-analyses and satellite-derived data sets" by Trolliet et al.

This manuscript compares hourly estimates of downwelling solar irradiance at the surface (DSIS) from two atmospheric reanalyses and three satellite-based products to measurements from PIRATA moorings at five locations in the tropical Atlantic. The results show that the satellite-based estimates generally perform much better than the reanalyses in terms of total DSIS and cloud forcing. These results will be useful for the research community and should be published. However, some revisions are necessary to improve the clarity and organization of the presentation. The manuscript is also presently a somewhat repetitive description of the statistics for each data set without much comparison between them or discussion of the reasons for the different or similar results for different data sets. Below are more detailed comments and suggestions for improvement.

Main comments:

The discussions of the results for the satellite-based data sets are very similar, and the performances relative to PIRATA are also similar. I suggest putting the satellite-based data set results in one section and discussing them together in order to avoid repetition. The MERRA and ERA results could also be put into the same section and discussed together for the same reason. Making these changes would improve the readability of the manuscript.

The layout of the figures can be improved. Currently they are structured so that a certain parameter is shown for all data sets in a given figure. However, in the text the results are discussed separately for each data set. It makes more sense to put all HelioClim plots (i.e. Fig. 2a,b, Fig. 3a,b, etc.) in the same figure, and the same for SARAH, CAMS, and the reanalyses.

As stated in the manuscript, potential biases in the PIRATA time series are an issue and complicate validation of the satellite-based data sets. These biases are discussed in section 1.1, but there's no summary or estimate of the overall uncertainty in the PIRATA hourly data. Can you provide an estimate? I would expect the buoy measurements may be biased low regardless of any aerosol buildup, based on the persistent low biases shown in Fig. 7 of Foltz et al. (2013), possibly due to fading of the radiometers' coatings with time. It might be helpful to plot the DSIS bias as a function of DSIS to help figure

out if biases of buoy DSIS may be partially to blame. I would expect the bias may be larger for larger DSIS if the buoy data have biases, essentially due to a bias in the buoy radiometers' gain coefficients. I don't see any evidence of this dependence in your figures, but it's difficult to tell for sure.

Other comments:

It's unclear how repetitive buoy tilting/rocking from waves would introduce a mean bias for a daily average (p. 3, lines 21-24). A brief explanation here would help. For a systematic tilt (e.g. on the equator due to strong zonal currents) it's easier to imagine.

p. 4, line 19: I would expect equatorial moorings to be influenced the most by tilt due to currents. North of about 8N currents should be much weaker in the mean.

Foltz et al. (2013) found a significant low bias from the mooring at 19S,34W despite no apparent dust buildup. It's not clear why, but it could explain your large discrepancies at that location.

p. 5, lines 21-22: Why not use the same EO as is used for PIRATA? That would ensure that differences in DSIS are the only thing contributing to differences in KT. Or if the EO values from different data sets are basically the same, that should be stated.

p. 6, lines 4-5: It's not clear how 30-min values were converted to hourly. Do you add anomalies from the TOA irradiance to 1-min TAO irradiance, then average this to an hourly average?

The portion of section 2 on p. 7-8 describes methodology more than results, so could be moved to section 1.

Why do you show only the 6S, 10W location in the figures? Please explain. In Fig. 2 the font within the figure (Mean, bias, st-dev, corr_coeff) is too small to read.

p. 10, lines 15-20: Are you saying here that HelioClim does not have enough cloud radiative forcing? It seems like it, but not sure.

p. 10, line 23: Are the results for 0n0e and 0n10w shown in a figure or table?

p. 10, lines 25-26: Why the underestimation and overestimation? Low cloudiness?

p. 11, line 1: Are you referring to the bias for KT=0.7?

p. 11, line 13: Please explain why it is important that spatial gradients are reproduced.

p. 11, line 19: I don't see this underestimation ion Fig. 2c.

p. 12, line 27: What is special about KT=0.6-0.7 that results in large biases in the satellite analyses? Because it appears so consistently, it would be worthwhile to know.

p. 13, line 5: "do not correlate" might be too strong of a statement, since some correlations are 0.82-0.91.

p. 13, line 9: What is the difference between true solar time and mean solar time?

p. 13, line 17: MERRA results are in Fig. 5c,d according to figure caption, not Fig. 5a.

p. 14, line 20: This statement is very confusing.

---

## Referee Comment (RC2) · Anonymous Referee #2 · 15 Feb 2018

Review of "Estimating downwelling solar irradiance at the surface of the tropical Atlantic Ocean: A comparison of PIRATA measurements against several re-analyses and satellite-derived data sets" by Trolliet et al.

This manuscript compares estimates of downward surface solar irradiance (DSIS) from two reanalyses datasets and three satellite-based products to measurements from PIRATA moored buoys at five locations in the tropical Atlantic and shows that the satellite

data outperforms the reanalysis datasets. The results are very useful and should be published. However, I think the paper needs improvement before being fit for publication. As a reader, it's quite difficult to read it as it is currently laid out as you have many repetitive sections on the various datasets. Going back and forth between the various plots is also a bit cumbersome. I suggest restructuring the paper to make it easier to read. Key results are also hidden in the middle of the paragraphs of repetitive text. I would also suggest that the main results and what makes this study different to the ones you have referenced widely are made clear for the reader.

Some comments (see attachment for some language suggestions also):

- Perhaps give the buoys names other than their lat/lon coordinates? - State why you used TOA on page 5 line 4. - I'd suggest a table for the datasets described in sections 1.2 -1.7 The text in these sections is repetitive and could be tightened. - You've looked at cloud cover – could you consider using integrated cloud condensate instead as it provides more information? - Tables 2-4 – fix caption. Some plots of these values would be more useful and may highlight the trends better. - Results section should indicate what you've plotted before being discussed in the next section. - Ensure all figures included are discussed – if not, remove unmentioned ones. - Discussion section needs restructuring to eliminate the repetition. - Also ensure key results are clear. - Alot of your results are consistent with other studies – please highlight the novel aspects of your study. - Can you do further analysis to investigate exactly why the reanalysis products are worse. - Make it clear why you include hourly and daily results. - Figure 3 – a different scale would mean less white space. Also looks too digitised. - Figure 4 – refine the scale used. - Figure 5 – contour bar too long relative to the plots

Please also note the supplement to this comment:
https://www.ocean-sci-discuss.net/os-2017-95/os-2017-95-RC2-supplement.zip

---

## Author Comment (AC1) · 22 Mar 2018

The attached document comprises a lot of editorial and language suggestions which highly improve the quality of the paper. I deeply thank the referee for these comments. I have taken all of them into account into the document. Going back and forth between the various plots is also a bit cumbersome.

»» The different figures have been moved and are now following the corresponding

sections. The plots have been clustered in two sets of figures: one for the satellite-derived data sets and one for the re-analyses. I suggest restructuring the paper to make it easier to read. Key results are also hidden in the middle of the paragraphs of repetitive text.

»» Thank you for this comment. We fully agree. Discussion and results sections have been restructured to improve the clarity and organization of the presentation and to avoid repetition in the description of the statistics for each data set. I would also suggest that the main results and what makes this study different to the ones you have referenced widely are made clear for the reader.

»» The text have been revised in the discussion part in order to make clear for the reader what makes this study different to the ones referenced and the innovation.

Perhaps give the buoys names other than their lat/lon coordinates?

»» As far as we know, the PIRATA buoys appellation is not an official one. In all the document provided on the PIRATA website (https://www.pmel.noaa.gov/gtmba/pirata), the buoys are named by their lat,lon. To avoid confusion, we have decided to keep this appellation, even though it is unusual.

State why you used TOA on page 5 line 4.

»» The paragraph following table 1, line 6-10 p5 and 1-7 p6 has been modified in order to highlight the use of KT index, and so on TOA.

I'd suggest a table for the datasets described in sections 1.2

»» Thank you, we have added a table (Table 2) which describes the characteristics of the five data sets.

1.2-1.7 The text in these sections is repetitive and could be tightened.

»» Sections have been reshaped in order to avoid repetition. The three satellite data sets have been brought together in section 1.2. The two re-analysis data sets have

been brought together in section 1.3.

You've looked at cloud cover – could you consider using integrated cloud condensate instead as it provides more information?

»» The cloud cover is a meteorological variable expressed in okta. We have not looked at the cloud cover because we do not have this information. CRS data set is providing the fraction of pixel covered by cloud, usually called cloud coverage, which is not the cloud cover. The text has been modified to make it clear, l20, p7. CRS data set does not provide the water/ice content.

Tables 2-4 – fix caption. Some plots of these values would be more useful and may highlight the trends better.

»» The different tables have been re-structured following the recommendations of the reviewers in order to make to improve the clarity of the content. We have combined the satellite-derived data sets on the one hand and re-analysis data sets on the other hand to better see the trends.

Results section should indicate what you've plotted before being discussed in the next section. – Ensure all figures included are discussed – if not, remove unmentioned ones. Discussion section needs restructuring to eliminate the repetition. - Also ensure key results are clear.

»» The sections "Results" and "Discussion" have been rewritten and we have taken these comments into account.

A lot of your results are consistent with other studies – please highlight the novel aspects of your study.

»» As already mentioned above, the text have been revised in the discussion part in order to make clear for the reader what makes this study different to the ones referenced and the innovation.

Can you do further analysis to investigate exactly why the reanalysis products are worse.

»» We are not knowledgeable enough to do such a new investigation. Other researchers are investigating these aspects.

Make it clear why you include hourly and daily results.

»» A paragraph has been added in Section 1.8 explaining that the study has been conducted on daily values for several reasons. One reason is that the performances may differ across these different time-scales. Another reason is that the daily values are the basis for constructed the monthly and yearly means, which are used in climatology. In addition, dealing with daily values allows comparing our results to already published works as it will be seen in Section "Discussion".

Figure 3 – a different scale would mean less white space. Also looks too digitised.

»» The figure has been regenerated taking into account this comment.

Figure 4 – refine the scale used.

»» The figure has been regenerated taking into account this comment.

Figure 5 – contour bar too long relative to the plots

»» The figure has been regenerated taking into account this comment.

---

## Author Comment (AC2) · 6 Apr 2018

Main comments: The discussions of the results for the satellite-based data sets are very similar, and the performances relative to PIRATA are also similar. I suggest putting the satellite-based data set results in one section and discussing them together in order to avoid repetition. The MERRA and ERA results could also be put into the same section and discussed together for the same reason. Making these changes would

improve the readability of the manuscript.

»» Thank you for this comment. We fully agree. Discussion and results sections have been restructured to improve the clarity and organization of the presentation and to avoid repetition in the description of the statistics for each data set. The three satellite-derived data sets are now present and discussed together, as the two re-analysis data sets. The layout of the figures can be improved. Currently they are structured so that a certain parameter is shown for all data sets in a given figure. However, in the text the results are discussed separately for each data set. It makes more sense to put all HelioClim plots (i.e. Fig. 2a,b, Fig. 3a,b, etc.) in the same figure, and the same for SARAH, CAMS, and the reanalyses.

»» The different figures have been moved and are now following the corresponding sections. The plots have been clustered in two sets of figures: one for the satellite-derived data sets and one for the re-analyses. As stated in the manuscript, potential biases in the PIRATA time series are an issue and complicate validation of the satellite-based data sets. These biases are discussed in section 1.1, but there's no summary or estimate of the overall uncertainty in the PIRATA hourly data. Can you provide an estimate?

»» You are right; these potential biases are an issue. According to the literature we have cited, the uncertainties are complex to model because of the large numbers influencing factors that occur with different time scales. For example, the wave effect and the aerosol effect do not influence the buoy at the same scale. These influencing factors are difficult to assess. Accordingly, we are not able to provide an estimate for the overall uncertainty in the PIRATA hourly data. The text has been modified and these difficulties have been underlined in the conclusion.

I would expect the buoy measurements may be biased low regardless of any aerosol buildup, based on the persistent low biases shown in Fig. 7 of Foltz et al. (2013), possibly due to fading of the radiometers' coatings with time. It might be helpful to plot

the DSIS bias as a function of DSIS to help figure out if biases of buoy DSIS may be partially to blame. I would expect the bias may be larger for larger DSIS if the buoy data have biases, essentially due to a bias in the buoy radiometers' gain coefficients. I don't see any evidence of this dependence in your figures, but it's difficult to tell for sure.

»»The presented graphs in Section 3 may help to answer these concerns though it is difficult to tell for sure as written by the Reviewer. The 2D histograms as well as the comparison of monthly means may provide insight of the possible relationships between the errors and the DSIS. No clear relationship emerges, and it depends upon the data set. For example, one may see in the comparison of the monthly means for satellite-derived data sets that the bias is greater for medium DSIS occurring in November and December and not for the greatest ones. Other comments: It's unclear how repetitive buoy tilting/rocking from waves would introduce a mean bias for a daily average (p. 3, lines 21-24). A brief explanation here would help. For a systematic tilt (e.g. on the equator due to strong zonal currents) it's easier to imagine.

»» Katsaros and DeVault (1986) distinguished two main kinds of errors: the errors due to rocking motion caused by waves and the errors due to a mean tilt. As you mentioned, the last one is easier to imagine. The first one can be approached by the two following extreme cases: (i) the buoy motion is in the direction of the sun and (ii) the buoy motion is perpendicular to that direction. In the first situation, Katsaros and DeVault (1986) expressed the error in irradiance measurement as a combination of losses produced by a motion away from the sun and gains by the tilting of the buoy toward the sun. By means of an analytical model and gross assumptions, Katsaros and DeVault (1986) concluded that "the average error for a cycle of motion will not be zero but will not be large". In the second situation, the effect of a perpendicular movement is always a loss, due to the loss of the sky portion seen by the pyranometer. Katsaros and DeVault (1986) calculated that the loss is of the order of 10% in hourly mean of irradiance for $10°$ tilt and solar zenithal angle greater than $30°$. For daily averaging periods, the

influence of the buoy movement is a combination of the two cases. As a consequence, compensating errors would often lead to smaller errors in measurement of daily means of irradiance. The text has been modified to bring this explanation as requested. p. 4, line 19: I would expect equatorial moorings to be influenced the most by tilt due to currents. North of about 8N currents should be much weaker in the mean.

»» Yes you are right, the mooring located in the equatorial band are subjected to tilt due to current. Foltz et al. (2013) wrote: "Errors due to buoy tilt are difficult to quantify (MacWhorter and Weller 1991), but are likely to be significant only at locations with strong mean currents (i.e., in the strong westward flow along the equator and eastward flow between 4 and 8N in the tropical Atlantic)." Hence, these stations were excluded in the study. North of 8 °N, "tilt biases are not expected to be significant in the 12–21N latitude band, where monthly-mean current speeds are ,20 cm s-1" (Foltz et al., 2013). The stations in this latitude band were excluded because of the contamination by African dust (Foltz et al., 2013). The text has been modified to make it clearer. p. 5, lines 21-22: Why not use the same EO as is used for PIRATA? That would ensure that differences in DSIS are the only thing contributing to differences in KT. Or if the EO values from different data sets are basically the same, that should be stated.

»» Each data set uses its own EO, which is provided by external astronomical models. The innovation of satellite-derived model and re-analyses is in the modelling of the clearness index KT, and not in E0. If we use the same E0 for all data sets, including PIRATA, this would create artificial distortions. This is why we use E0 of each data set. In any case, the E0 differs slightly from a data set to another by a few W m-2, excepted for MERRA-2 as shown in the study (mean and true solar time). The text has been modified to make it clearer. p. 6, lines 4-5: It's not clear how 30-min values were converted to hourly. Do you add anomalies from the TOA irradiance to 1-min TAO irradiance, then average this to an hourly average?

»» Regarding SARAH-2, the instantaneous values every 30 min were converted into instantaneous clearness index every 30 min. Assuming that KT is constant over 30

min, each instantaneous KT is multiplied by the corresponding E0 integrated over 30 min, yielding 30 min irradiation. These 30 min irradiations are summed two by two to yield hourly irradiations, and then hourly means of irradiance. The text has been modified in order to add this information. The portion of section 2 on p. 7-8 describes methodology more than results, so could be moved to section 1.

»» The section "Results" has been split and this part of the manuscript is now in the Section 1.

Why do you show only the 6S, 10W location in the figures? Please explain.

»» The station 6s10w has been chosen as an illustrative example. This precision has been added in the text. The plots for the others locations have been added in appendix in order to guarantee the ability to the reader to compare the different results.

In Fig. 2 the font within the figure (Mean, bias, st-dev, corr_coeff) is too small to read.

»» This has been corrected taking into account this comment.

p. 10, lines 15-20: Are you saying here that HelioClim does not have enough cloud radiative forcing? It seems like it, but not sure.

»» You are right, these sentences were unclear. We have rewritten this part.

p. 10, line 23: Are the results for 0n0e and 0n10w shown in a figure or table?

»» The different plots for the five buoys are now provided in Appendix.

p. 10, lines 25-26: Why the underestimation and overestimation? Low cloudiness?

»» answer

p. 11, line 1: Are you referring to the bias for KT=0.7?

»» This comment has been taken into account during the rewriting of the section.

p. 11, line 13: Please explain why it is important that spatial gradients are reproduced.

»» answer

p. 11, line 19: I don't see this underestimation ion Fig. 2c.

»» Indeed, the underestimation is only visible on the frequency histogram. This comment has been taken into account in the rewriting of the analysis of the frequency distribution of PIRATA measurements and satellite-derived data sets.

p. 12, line 27: What is special about KT=0.6-0.7 that results in large biases in the satellite analyses? Because it appears so consistently, it would be worthwhile to know.

»» A sentence suggesting explanations of the large over-estimation in the satellite-derived data sets has been added in the description of the frequency distribution of PIRATA measurements and satellite-derived data sets. TO DO

p. 13, line 5: "do not correlate" might be too strong of a statement, since some correlations are 0.82-0.91.

»» Thank you for the comment, it was a mistake. The text has been corrected from "do not correlate" to "correlate".

p. 13, line 9: What is the difference between true solar time and mean solar time?

»» answer The mean solar time corresponds to the time defined through the time duration for one earth rotation divided into 24 h as an average. The sun is approximately at its zenith when the mean solar time is equal to 12 h. Consequently, the mean solar time is not the same everywhere on the earth. It depends upon the longitude. The true solar time takes into account that the earth's angular speed varies slightly throughout the year because of the elliptic orbit of the earth. Combined with the rotation of the earth on itself, which is very regular, it results that the sun does not reach its highest position in the sky at 12 h mean solar time every day. In other words, the true solar time corresponds to the time determined every day by the actual position of the sun in the sky. The true solar time is that needed for computing the solar zenithal angle accurately enough. This angle intervenes twice: firstly to compute the irradiance impinging

on the horizontal plane at the top of atmosphere and secondly as a major input to the radiative transfer model. Hence, an error in this angle yields an error in the estimated DSIS. The mean solar time can differ up to 17 min from the true solar time. These precisions have been added in Section "Discussion and results".

p. 13, line 17: MERRA results are in Fig. 5c,d according to figure caption, not Fig. 5a.

»» You are right, the figures has been mixed. It has been corrected.

p. 14, line 20: This statement is very confusing.

»» The paragraph has been rewritten in order to avoid confusion. The observations are now in the Section "Daily analyses of the re-analysis data sets".

---

## Author Response (AR2)

**Letter to the Editor**

Dear Dr Simon Josey,

I thank you again for all the efforts you are making in handling this manuscript. The two reviewers have considered the revised version of the manuscript much easier to read and the results easier to interpret. Both of them suggested minor comments that mostly consist in edition and language suggestions. All suggestions have been taken into account in this 2$^{nd}$ revision. In addition, this 2$^{nd}$ revision integrates the few technical points highlighted by the Reviewer #2. Details are given in the following pages.

As requested, the manuscript was revised by a fluent English speaker. This leads to rewriting some parts of the text, including the abstract. Following suggestion by reviewer #2, the title has been changed.

While the revised manuscript was under review, we found a small mistake in our computations. It has a small to moderate impact when looking at the final. However, it was worth checking and redoing the calculations to ensure that we were discussing the correct numbers. The 2$^{nd}$ version comprises the new numbers.

The different tables of results with and without the correction are reported at the end of this letter for your information. In many cases, the original numbers are untouched. The differences are at the level of the last digit and amounts typically to 1 or 2. The most noticeable changes are for the correlation coefficients which have been slightly increased or decreased depending on the case.

In Table 3, the differences for the correlation coefficient are less than 0.005 in absolute values. Changes up to 0.01 are found for the moorings 0n0e and 6s10w for the clearness index (numbers in italics) for the HC3v5 and SARAH-2 data sets. In the same Table, the only other noticeable change is for the bias at the mooring 0n10w and the HC3v5 data set with an increase in bias of 8 W/m² (3 %). There is no change for the standard deviation, except for the last digit (1 or 2).

In Table 4, the differences for the correlation coefficient are often less than 0.007 in absolute values. Changes up to 0.019 are found for the moorings 0n0e for the three data sets. In the same Table, the only other noticeable change is for the bias at the mooring 0n10w and the HC3v5 data set.

In Table 5, the differences for the correlation coefficient are less than 0.007 in absolute values for ERA5. More noticeable changes (coefficients less than before) may be found for MERRA-2 at moorings 0n10w, 0n23w and 19s34w for the clearness index (numbers in italics).

In Table 6, the correlation coefficients are less than before for MERRA-2 and ERA5. This enhances the conclusions on these data sets.

These corrections have little to no influence on the conclusions of the study. The only modification is found for the CRS data set whose performances are now closer to the HC3v5 ones. As a consequence, we have rewritten a part of the Section Results in order to underline this new result as well as the Abstract. From our point of view, these minor modifications do not change the structure or the main message of the article but we felt that they have to be clearly mentioned to you.

Best regards,

Mélodie Trolliet

**Answer to anonymous Referee #1**

My only suggestion is that the language in the paper is thoroughly checked by a native English speaker.

>>>> The manuscript has been kindly checked by a fluent English speaker. Several parts of the text, including the abstract, have been rewritten.

**Answer to anonymous Referee #2**

The title is rather long and could be shortened a little bit to "Downwelling surface solar irradiance in the tropical Atlantic Ocean: A comparison of PIRATA measurements to reanalyses and satellite-derived data sets"

>>>> The title has been modified as suggested. Thank you!

p. 2, line 10: Probably better to replace "hardly known" with something like "not well known"
p. 2, line 27: Replace "effort is needed" with "efforts are needed"
p. 3, line 5: Replace "comprises" with "consists of" and "atlas" with "ATLAS"
p. 4, line 6: Replace "wind currents" with "winds"
p. 8, line 6: Change "on" to "to"
p. 28, line 7: Change "bias" to "biases"
p. 29, line 29: Change "than" to "as"

>>>> All these suggestions have been integrated in the manuscript. Thank you for the careful reading.

p. 3, line 9: "Sensors are cleaned manually every trimester" I'm not sure what this means. My understanding is that they are replaced with clean sensors during every servicing cruise and that they are never cleaned while deployed on the buoys.

>>>> The sentence has been rephrased in order to avoid misunderstanding in the maintenance protocol. Text is now:

"The sensors are deployed for about one year on average before replacement. Sensors are replaced with clean sensors during every yearly servicing cruise"

p. 4, lines 25-26: I don't think Foltz et al. (2013) recommended discarding the data from those buoys, just that it may be more difficult to attribute biases to dust because of possible tilt biases. Maybe best to delete this statement.

>>>> The statement has been deleted in order to avoid misinterpretation of the Foltz's work.

I'm curious why KT is used for clearness index instead of CI. Is KT a commonly used abbreviation? Same thing for using E for DSIS.

>>>> The use of KT in order to define the clearness index is a shared practice in the radiation field for several decades. As far as I know, it comes from German language. Same thing concerning the use of E, it corresponds to the classical terminology.

Figure 1: You could plot annual mean DSIS from one of the satellite products as the background instead of solid blue. That would help to visualize the different regimes the buoys are in.

>>>> Thank you for this comment. We have made some experiments in order to visualize the different regimes the buoys are in. Plotting the annual mean DSIS from one of the satellite products provide information on this point. However, it also raises some interesting questions. As a consequence, we have decided to keep the message easy on this point for this article, and to develop this question in a new study. We are currently working on an article under revision in the journal ASR, dealing with the ability of the PIRATA buoys to describe the different irradiance regimes the buoys are in.

p. 15, line 24: "Sara" Do you mean "SARAH-2"?

>>>> Indeed, it is SARAH-2. The mistake has been corrected.

I found it a little strange to have the discussion of daily averaged results after the discussion of monthly mean climatologies. It's a minor thing, but it might make more sense to go from hourly means to daily means to monthly means.

>>>> The study does not provide any information on monthly mean climatologies. The plots shown in the study corresponds to the monthly means of the hourly DSIS. They are presented here in order to check if the performances are similar for any month. No detailed study is done on monthly means. Our graphs were drawn with the available data that are common to all data sets. This does not ensure that we have enough data to compute the actual monthly means of the DSIS. Following the recommendations of the World Meteorological Organisation, a monthly mean should be computed with a maximum of 5 missing days. Here, this constraint has not been followed because the aim was to check the consistency of the performances of a data set from month to month. In order to avoid confusion between computation methods, we have decided to keep the structure as it is.

p. 26, lines 25-27: This sentence is very confusing.

>>>> The sentence has been rephrased in order to avoid confusion as follows :

[revised manuscript text omitted]